# The Golgi checkpoint: Golgi unlinking during G2 is necessary for spindle formation and cytokinesis

Fabiola Mascanzoni, Inmaculada Ayala, Roberta Iannitti, Alberto Luini, Antonino Colanzi

**Entry into mitosis requires not only correct DNA replication but also extensive cell reorganization, including the separation of the Golgi ribbon into isolated stacks. To understand the significance of pre-mitotic Golgi reorganization, we devised a strategy to first block Golgi segregation, with the consequent G2-arrest, and then force entry into mitosis. We found that the cells forced to enter mitosis with an intact Golgi ribbon showed remarkable cell division defects, including spindle multipolarity and binucleation. The spindle defects were caused by reduced levels at the centrosome of the kinase Aurora-A, a pivotal spindle formation regulator controlled by Golgi segregation. Overexpression of Aurora-A rescued spindle formation, indicating a crucial role of the Golgi-dependent recruitment of Aurora-A at the centrosome. Thus, our results reveal that alterations of the pre-mitotic Golgi segregation in G2 have profound consequences on the fidelity of later mitotic processes and represent potential risk factors for cell transformation and cancer development.**

## Introduction

The G2/M transition is a critical preparatory step for the equal separation of genetic material between the daughter cells, which is crucial for cell physiology. The mitosis onset is under the surveillance of checkpoints, which are signaling pathways that arrest cell cycle progression in G2 in the presence of incomplete DNA replication or DNA damage (1). Once the checkpoints are satisfied, the irreversible commitment to mitotic entry is triggered by the activation of cyclin-dependent kinase 1 (CDK1), which induces rapid and profound modifications of cell shape, involving gradual reorganization from a flat to spherical geometry, accompanied by centrosomes (CEs) separation, nuclear envelope breakdown, chromatin condensation, and spindle formation (2).

In recent years, it has emerged that the control of DNA replication is not the only factor governing entry into mitosis, which also requires additional and crucial preparatory steps that occur during G2 (2, 3). Indeed, as the cells approach mitosis, the G2-restricted

expression of the scaffold protein DEPDC1B and increased levels of inactive CDK1 induce the selective disassembly of focal adhesions (FAs) (4, 5). FAs are integrin-based macromolecular complexes that link the actin cytoskeleton to the extracellular matrix (6). FA dismantling allows the retraction of the cell margin and cellular rounding by actin-based remodeling processes. Incorrect mitotic rounding alters spindle morphology and chromosome segregation (7). FA dismantling is important as its block impairs entry into mitosis (4).

An additional preparatory step involves the Golgi complex (GC). The GC has a central role in the secretory pathway (8, 9) and is generally organized as polarized stacks of cisternae connected by lateral tubules, forming a structure known as the "Golgi ribbon" (10). We and others have previously shown that during G2, the GC is unlinked into stacks and that the block of this unlinking process causes a potent G2-arrest (11, 12), indicating that a mitotic "Golgi checkpoint" ensures that cells do not enter into mitosis in the presence of an intact ribbon (12, 13). The GC is the only organelle to show a structural modification in G2 as a prerequisite for entry into mitosis (10). More recently, we have also revealed the basic elements of the Golgi checkpoint. Indeed, we have found that during G2, the GC unlinking into isolated stacks acts as a signal to induce Src activation at the trans-Golgi network (14). Importantly, Src activation can also be stimulated in interphase cells by drug-induced unlinking, in line with the evidence that many signaling pathways are located at the GC and are modulated by its architecture to regulate high-order functions (15). Moreover, we have shown that the Golgi-activated Src phosphorylates Tyr148 of the mitotic kinase Aurora-A, stimulating its recruitment at the CEs and the kinase activity (14). This residue is a novel and Golgi-specific mechanism of Aurora-A activation that is necessary for triggering mitotic entry through CDK1 activation (16). Aurora-A activation at the CEs is complex, also requiring the redistribution of a set of proteins from the FAs to the CEs (17). Thus, Aurora-A acts as an integrator at the CEs of stimuli from the separated GC ribbon and the dismantled FAs, suggesting that these processes are functionally coordinated to control important mitotic events (2).

After the mitosis onset, CDK1 (18, 19) triggers the disassembly of the stacks into vesicular/tubular clusters dispersed in the cytoplasm (20, 21, 22). Also, this step is important for mitotic progression

Institute of Experimental Endocrinology and Oncology "G. Salvatore" (IEOS), National Research Council (CNR), Naples, Italy

Correspondence: a.luini@ieos.cnr.it; a.colanzi@ieos.cnr.it

as inhibition of stack disassembly after the mitosis onset results in spindle multipolarity and metaphase arrest (23). Finally, during telophase, the Golgi clusters gradually reassemble into new Golgi ribbons in each daughter cell (24, 25).

Despite the progress in understanding the mechanisms that coordinate the mitotic disassembly of the GC with the regulation of cell division, some crucial questions remain unaddressed.

One of these open questions is the understanding of the physiological role of Golgi ribbon separation into isolated stacks during G2. Therefore, we addressed the function of the G2-specific GC segregation on the fidelity of the division process. The GC architecture is maintained by multiple factors, including cytoskeleton (15, 26) and structural proteins, such as the Golgi reassembly and stacking proteins (GRASPs) (27) and members of the golgin family (28), which act as membrane-tethering proteins (10). The ribbon organization is characterized by a significant structural plasticity, thanks to a dynamic equilibrium between the formation and cleavage of the inter-stack tubular connections (10, 11). Ribbon formation is driven by the Golgi "matrix" proteins GRASP65 and GRASP55 (27), which direct the tethering and fusion of newly formed tubules with adjacent stacks (10). In contrast, the cleavage of these tubules is operated by the fission-inducing protein BARS/CTBP1-S (11). The ribbon/stacks equilibrium is regulated by phosphorylation events. More in detail, during G2, a complex signaling converging on the kinases JNK2 (29), PLK1 (30, 31), and MEK/ERK (32, 33) induces the phosphorylation of GRASP65 and GRASP55, inhibiting their pro-ribbon role (10), resulting in GC unlinking (11).

Among the GRASPs, our previous studies have revealed a key role for GRASP65 because in the unphosphorylated state, in addition to promoting the formation of membrane continuities connecting the stacks, it also induces the stabilization of Golgi-associated microtubules (MTs), leading to their acetylation, which drives the clustering of Golgi stacks, thus facilitating ribbon formation (34). GC ribbon and microtubule stabilization are both inhibited by JNK-mediated phosphorylation of Ser274 of GRASP65 (29, 34).

Thus, exploiting the knowledge gained about the regulation of GC structure, we developed the strategy to first accumulate cells in G2 with an intact Golgi ribbon through JNK inhibition (29, 34) and then force them to enter mitosis by a drug-induced activation of CDK1 (35). We found that the cells showed profound alterations of fundamental mitotic processes, including the formation of multipolar spindles and binucleation. Spindle alterations were not caused by a drag force exerted on the separated CEs by an intact ribbon as after the mitosis onset, the GC was disassembled also in cells treated with JNK inhibitors. The cells that entered mitosis with an intact Golgi ribbon showed reduced levels of Aurora-A at the CEs. Importantly, Aurora-A levels and correct spindle formation were recovered by Aurora-A overexpression or by inducing Golgi unlinking or disassembly, indicating that these defects were the direct consequence of the block of the GC unlinking in G2. Overall, our data revealed that the GC-originated Src-Aurora-A signaling that is triggered by ribbon unlinking in G2 is crucial for correct spindle formation in mitosis and cytokinesis and, therefore, for the preservation of the integrity of the genome. In conclusion, entry into mitosis with an intact Golgi ribbon can recapitulate in spindle

aberrations and tetraploidization, with important potential implications in the maintenance of tissue homeostasis or in the development of ageing and cancer (36, 37).

# Results

## Setting up an approach to force entry into mitosis with an intact Golgi ribbon

To investigate the role of GC unlinking in mitotic entry, we have developed a two-step strategy to first accumulate cells in G2 with an intact Golgi ribbon and then force them to enter mitosis. To block GC unlinking and investigate the downstream effects, we employed a well-characterized strategy involving the pharmacological inhibition of the phosphorylation of GRASP65, which induces a potent GC-dependent G2 arrest, coupled with artificial disassembly of the GC as a control to identify the direct consequences of the block of the GC unlinking (29).

To this purpose, we employed normal rat kidney (NRK) cells as they are noncancerous cells and generally used as a mitosis model because they are easy to synchronize. The cells were arrested at the beginning of the S-phase using the aphidicolin-block protocol (38) (Figs 1A and S1A and B). Then, aphidicolin was washed out, and during G2, the cells were treated for 2 h with DMSO as a control or with SP600125 (SP), which inhibits the JNK2-mediated phosphorylation of GRASP65, resulting in the formation of a continuous Golgi ribbon and G2 arrest (14, 29). 2 h after the addition of SP, part of the cells was also treated with MK1775 (MK), an inhibitor of Wee1, which is an inhibitor of the kinase CDK1 (35). This last treatment induces the activation of CDK1 and a rapid and synchronous entry into mitosis (35). At the end of the incubation with MK, the cells were fixed and stained for immunofluorescence with an anti-GRASP65 antibody and Hoechst to label the GC and the DNA, respectively. Confocal microscopy analysis of the cells in G2 showed that the GC of control cells appeared in the usual form of distinct groups of membranes localized mainly around the nucleus, whereas SP inhibited GC unlinking, as previously shown (Fig 1B) (29). The quantification of Golgi objects shows that the addition of MK had no effects on the Golgi structure of the cells in G2, whereas SP reduced its number by about 40%, in line with the effect of ribbon reformation. Furthermore, MK added to SP did not counteract the Golgi compaction effect induced by SP (Fig 1B and C). Next, we examined if MK treatment was able to force SP-treated cells to enter mitosis. Quantification of the mitotic index showed that SP induced about 75% inhibition of entry into mitosis compared with control cells, as previously shown (29). As expected, MK addition caused a 50% increase of the mitotic index in control cells and, more importantly, also fully rescued the SP-induced G2 arrest (Fig 1B and D). These results indicate that MK can override the Golgi checkpoint in cells with an intact GC ribbon in G2, indicating that the assay is suitable for testing whether the Golgi status in G2 (i.e., in the form of disassembled stacks as in controls cells or intact, in the case of SP-treated cells) affects the subsequent mitotic processes.

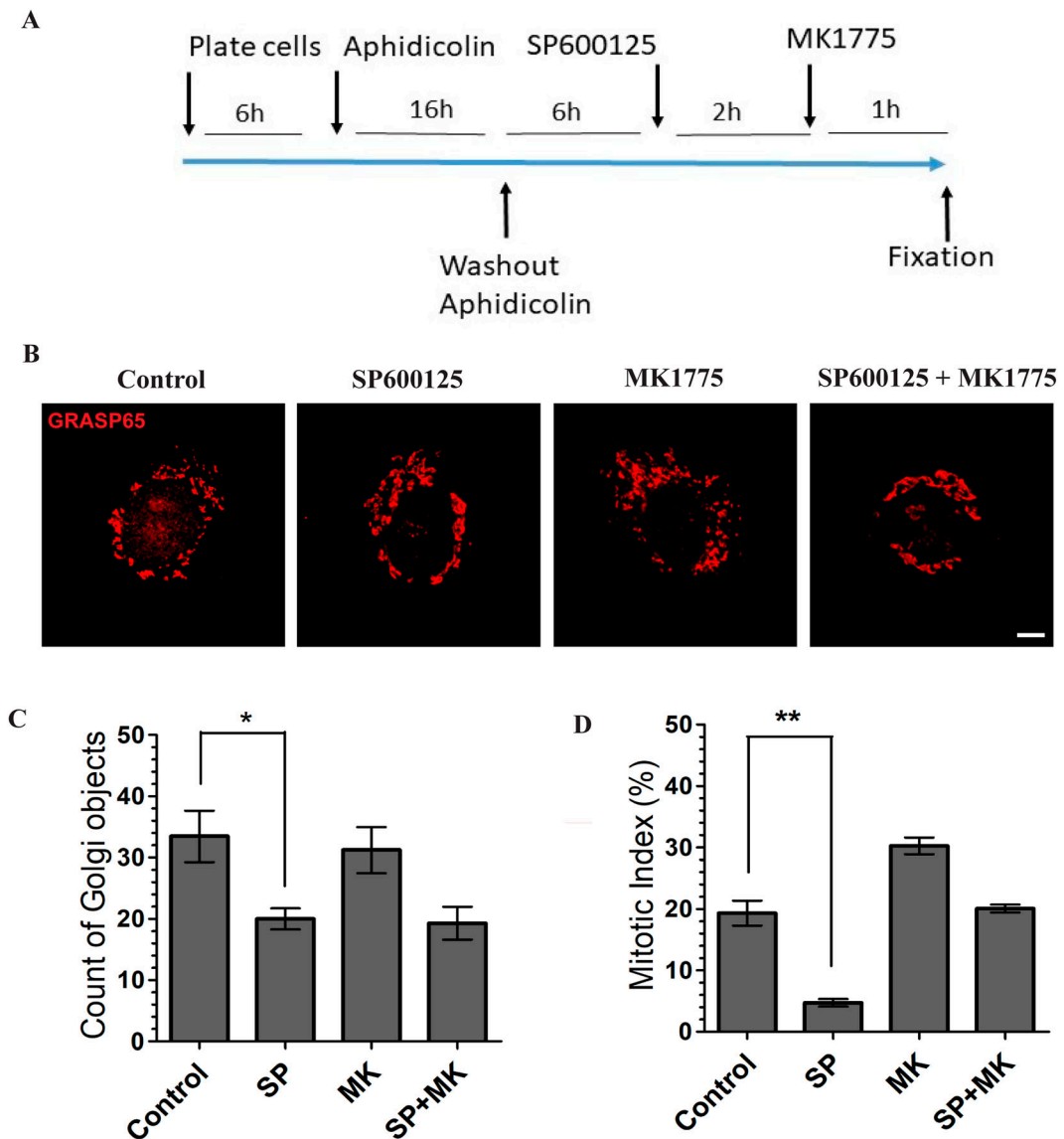

**Figure 1. Set up of experimental strategy to induce the cells for entry into mitosis with a compact Golgi ribbon.**
**(A)** Schematic description of the experimental protocol. Normal rat kidney (NRK) cells were grown on coverslips and arrested in S-phase with overnight treatment using aphidicolin. 6 h after aphidicolin washout, the cells were treated with 25 $\mu$M SP600125. 2 h later, 0.3 M MK1775 was added to the medium. The cells were further incubated for 1 h before fixation at the mitotic peak. **(B)** Representative confocal images of the G2 Golgi complex in NRK cells treated with vehicle (control) or the JNK inhibitor SP600125, in the presence or absence of MK1775. The Golgi complex was labeled with an anti-GRASP65 antibody. G2 cells were identified as described in (29). **(C)** Quantification of Golgi objects in NRK cells treated as in (A). Data are mean values (±s.d.) from three independent experiments. *$P < 0.0344$ (control versus cells with SP) ($t$ test). **(D)** Quantification of the mitotic index in NRK cells treated as in (A). More than 200 cells were counted for each condition. Data are means ± s.e.m. from three independent experiments; ** indicates $P < 0.005$, SP versus control (control) cells ($t$ test). Scale bar: 5 $\mu$m.

## Golgi unlinking during G2 is required for proper spindle formation

Next, we investigated the functional consequences of the forced entry into mitosis in cells with an intact GC in G2, focusing first on the spindle structure, whose formation is completed during metaphase. The major task of the spindle is to separate sister chromatids between daughter cells. Thus, this approach offers the opportunity to clearly identify potential defects (7). To visualize the spindle aberrations, the cells were forced to enter mitosis, as indicated above. Then, after fixation, they were labeled with antibodies against $\alpha$-tubulin and GRASP65 to identify the MTs and the GC, respectively, and Hoechst to visualize the DNA (Fig 2A and B). We found that 60% of the cells treated with SP and MK showed severe spindle alterations, whereas only a minor fraction of control cells, about 10%, showed alterations (Fig 2B and C), in line with the presence of transient defects normally detected also during unperturbed mitosis (39). The prevalent defect was multipolarity, with a minor fraction represented by disorganization of the spindle fibers (Fig S2) (39). Treatment with MK or SP alone did not significantly increase the frequency of spindle defects compared with

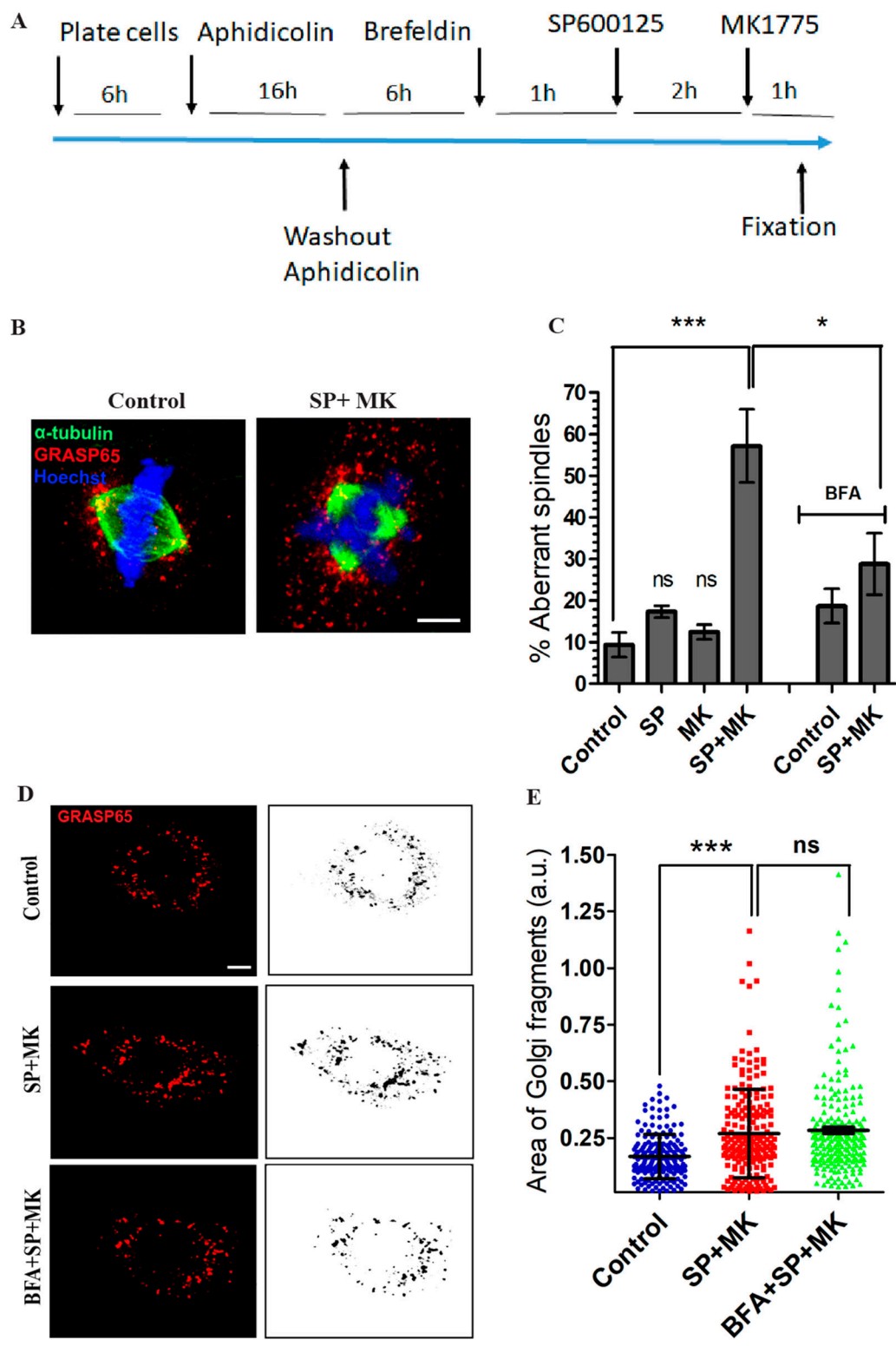

**Figure 2. Inhibition of mitotic Golgi unlinking causes spindles defects.**
**(A)** Schematic description of the experimental protocol. **(B)** Normal rat kidney (NRK) cells were treated as indicated in (A), fixed, and stained with antibodies against GRASP65 and α-tubulin to label the GC and microtubules, respectively, and with Hoechst 33342 to observe the DNA. Representative confocal images of NRK cells in metaphase treated with vehicle (control) or the combination of SP and MK, as indicated. Control cells showed a bipolar spindle, whereas the cells treated with SP and MK presented an aberrant spindle organization (tripolar). **(C)** Quantification of the spindle defects (percentage of aberrant spindles) observed in experiments performed as described in (A). At least 200 metaphase cells were counted for each experimental condition from three independent experiments. Scale bars: 5 μm. Data are means ± s.e.m. from three independent experiments. One-way ANOVA with Tukey's multiple comparison test. ***$P < 0.001$ (control versus SP + MK); *$P < 0.05$ (SP + MK versus SP + MK +

the control (Fig 2C). Therefore, to test whether the observed abnormalities were a direct consequence of the inhibition of GC unlinking, NRK cells were treated as above but with the difference that 1 h before addition of SP, part of the samples were also treated with brefeldin A (BFA) (Fig 2A) to induce artificial disruption of the GC (29, 40). BFA does not interfere with cell division and induces redistribution of Golgi enzymes into the endoplasmic reticulum (ER), whereas Golgi matrix proteins aggregate in the form of dispersed tubular vesicular clusters (41). Notably, the BFA-induced Golgi disassembly induced a 60% rescue of bipolar spindle formation events in cells treated with SP and MK (Fig 2C), indicating that a significant fraction of spindle defects directly resulted from an intact GC ribbon in G2 and not from untimely FA disassembly potentially caused by MK (5).

A previous study based on blocking stack disassembly through the formation of large diaminobenzidine polymers in the CG lumen has shown that inhibition of GC stacks disassembly in mitosis induces the formation of monopolar spindles, spindle assembly checkpoint activation, and metaphase arrest as a consequence of a steric hindrance of the intact Golgi stacks (23). Thus, a possible explanation of our results is that forcing entry into mitosis of G2 cells with an intact ribbon could affect spindle formation because of the persistence of large Golgi clusters and that BFA alleviated the defects by dissolving these clusters. To examine this hypothesis, we evaluated the extent of mitotic Golgi fragmentation by measuring the size of mitotic clusters in control cells or cells treated with SP and MK in the absence or presence of BFA. In control cells, the Golgi underwent extensive fragmentation, as expected. Similarly, in cells treated with SP and MK, the Golgi was also highly fragmented, albeit to a slightly lesser extent, as shown by the about 50% increase in the average area of the GC clusters (Fig 2D and E). The latter result is not surprising because even if SP inhibits GC unlinking, the activation of CDK1 after the mitosis onset stimulates Golgi disassembly pathways distinct from the JNK2-mediated GRASP65 phosphorylation (42). Importantly, the addition of BFA to cells treated with SP and MK did not alter the size distribution of the disassembled GC clusters compared with cells treated only with SP and MK (Fig 2D and E), excluding the possibility that BFA recovers spindle formation because of the reduction of the size of the clusters, suggesting that the spindle defects observed in metaphase are the direct consequence of the lack of Golgi unlinking during the G2 phase.

To test this conclusion by an independent approach, we induced constitutive and irreversible unlinking of GC into isolated stacks through the depletion of GRASP55 and examined whether this condition rescued spindle formation. Here, we employed TERT-RPE1 (RPE-1) cells as they are noncancerous cells that can be efficiently synchronized and also transfected. The cells were transfected with control non-targeting or GRASP55-specific siRNAs and synchronized using the double-thymidine protocol (29) (Fig S3). Constitutive GC unlinking is a condition that overrides the effect of SP on Golgi

compaction and G2 arrest (29). After thymidine washout, the cells were incubated in the absence or presence of SP and MK, fixed at the mitotic peak, and stained with antibodies against GRASP55 and α-tubulin (Fig S4). The quantification of Golgi objects shows that the addition of SP to control cells, even in the presence of MK, induces a reduction in Golgi objects, about 40%, as already shown for NRK cells. Instead, in cells depleted of GRASP55, the number of Golgi objects is increased by about three times, showing that the depletion of GRASP55 induces an irreversible unlinking of the Golgi structure. In these cells, the addition of SP and MK can still induce a minor reduction of Golgi objects, whose number remains about 40% higher than in control cells (Fig 3A). Importantly, the investigation of spindle morphology showed that in RPE-1 cells, the treatment with SP or MK alone induced an increase of aberrant spindles. However, the combination of SP and MK had a potent additive effect in cells treated with non-targeting siRNA (about 55% of cells with defective spindles, more than double with respect to SP or MK alone) but had a significantly lower impact in cells with the GC irreversibly unlinked by depletion of GRASP55 (GR-55 KD; about 30% of cells with defective spindles; Fig 3B). Thus, our data demonstrated that a significant fraction of the aberrant spindles observed in cells treated with SP and MK is the direct consequence of the lack of Golgi unlinking during G2.

### Golgi unlinking controls spindle formation through Aurora-A

Next, we addressed the question of why Golgi unlinking in G2 is necessary for spindle formation. The main organizers of the spindle are the CEs, which are duplicated during S-phase (43) and then separated during G2, in concomitance with ribbon unlinking (38, 44). The separated CEs reach the definitive localization at metaphase when they are located at the opposite poles of the cells. Starting from the G2 phase, the pericentriolar matrix (PCM) surrounding the CEs undergoes a profound modification. The PCM is a matrix composed of multiprotein complexes that control the polymerization and stability of three types of spindle fibers: the kinetochore MTs connecting the PMC to the kinetochores, the polar MTs that interconnect at the spindle midzone to push the spindle poles apart, and the astral MTs anchoring the PCM to the plasma membrane (43). This complex machine generates a combination of pulling and pushing forces on the chromosomes and the CEs to drive the formation of a symmetric and bioriented spindle, which is necessary for chromosome alignment at the metaphase plate (45).

The CEs are also physically connected to the GC through the polymerization of radial MTs, which form tracks along which Golgi membranes are transported to the cell center (46, 47). Therefore, it is possible that an intact Golgi ribbon could exert a drag force on the separating CEs during the G2/M transition, altering their position and thus resulting in aberrant spindle formation. To evaluate this possibility, we examined whether a bulky unseparated GC could

BFA); control versus BFA is not significant (ns) *P* = 0.2978 (ns); BFA versus BFA + SP + MK is not significant, *P* = 0.2750. **(D)** NRK cells were treated as indicated in (A), fixed, and stained with antibodies against GRASP65, to label the GC. Representative confocal images of the GC in NRK cells in metaphase treated with vehicle (control) or the combination of SP and MK in the absence or in the presence of BFA. For qualitative analysis of the Golgi, GRASP65 staining was processed using the "Invert" function of the ImageJ software package. Scale bar, 5 μm. **(E)** Quantification of the areas of GC clusters in control NRK cells or in cells treated with SP and MK, in the presence or absence of BFA. Data are mean values (±s.d.) from three independent experiments. ***P < 0.0001 (control versus cells with SP + MK) (*t* test); n.s., not significant (SP + MK versus BFA + SP + MK).

A

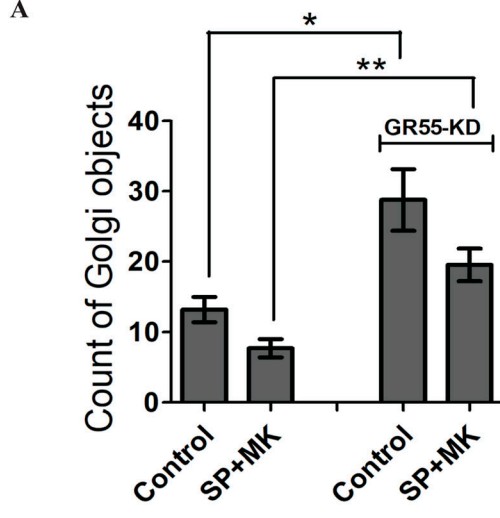

B

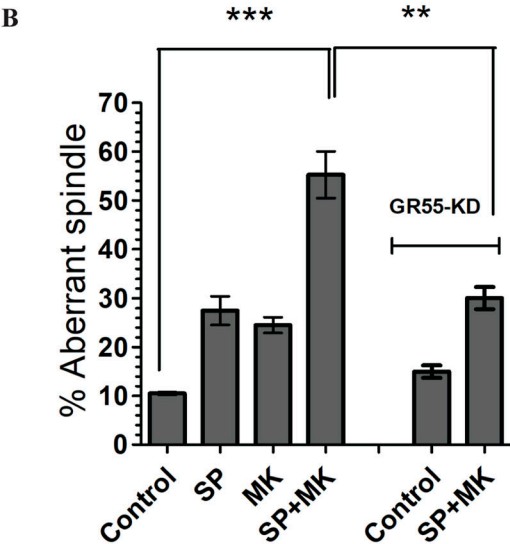

**Figure 3. GRASP55 depletion induces a significant rescue of spindle aberration.**
**(A)** Quantification of Golgi objects in RPE-1 cells, synchronized with thymidine double block, treated as in Fig 1A. Data are mean values (±s.d.) from three independent experiments. *P < 0.0311 (control versus GR-55 KD) and **P < 0.0032 (SP + MK versus GR55 KD + SP + MK). (t test). **(B)** Quantification of the spindle defects (percentage of aberrant spindles) observed in experiments performed in RPE-1 cells transfected with siRNA (non-targeting, control) or with siRNA against GRASP55 and treated as indicated. Data are means ± s.e.m. from three independent experiments. ***P < 0.001 (control versus SP + MK); **P < 0.01 (SP + MK versus GR55-KD + SP + MK) (t test).

perturb the correct separation and positioning of the CEs during G2 and prophase. To this purpose, HeLa cells were synchronized for G2/M transition by double-thymidine block and treated with vehicle or SP and MK, as shown in Fig 1A. The cells were then fixed and labeled with antibodies against pericentrin and GRASP65 to identify the CEs and the GC, respectively, and with Hoechst to monitor the level of chromosomes condensation (Fig 4A). To measure potential perturbations of CEs separation, we assessed their "centralization" by measuring the differential distance of the two CEs from the

nucleus center (Δ = delta centralization) (Fig 4B) and CE "separation" by measuring the distance between the two centrosomes (d = distance) (Fig 4C). Our results demonstrated that the block of Golgi unlinking did not significantly alter the positioning or the separation of the CEs during both G2 and prophase (Fig 4B and C), excluding the possibility that a bulky unlinked GC could exert an important drag force on CEs repositioning. Our data are also supported by the finding that the connection between the CEs and the GC is temporarily loosened during G2 (48).

Having excluded that an intact ribbon could hamper CE positioning in G2, we focused on the major Golgi-dependent spindle defect (i.e., multipolarity). Using confocal microscopy, we observed that this alteration is the consequence of the formation of multiple MT nucleation foci (Fig 5A) that, in turn, are the result of the fragmentation of either CEs or PCM (45). To this end, the cells treated with control buffer or with SP and MK were fixed and labeled with antibodies against ninein (49), which is a core CE marker that localizes primarily to the postmitotic mother centriole and pericentrin, a PCM component (50). As shown in Fig 5B, control cells showed two pericentrin foci that colocalized with two ninein-labeled structures. Conversely, the cells treated with SP and MK showed multiple pericentrin-labeled foci but only two ninein-marked CEs. Thus, these results demonstrated that lack of ribbon unlinking in G2 causes PCM fragmentation in metaphase, resulting in the aberrant spindle formation (45).

The fragmentation of PCM results from imbalanced forces exerted by the MT fibers on the PCMs. A major regulator of spindle fibers formation and function is the kinase Aurora-A, which during G2 becomes associated with the CEs to prepare the PCM to control MT nucleation and stabilization. The PCM comprises many scaffold proteins forming a large multimeric structure that anchors the newly nucleated MTs (51). Significantly, reduced Aurora-A activity results in altered MT nucleation and stabilization, leading to unbalanced MT-mediated forces during prometaphase and PCM fragmentation (51). Of note, ribbon unlinking acts as a specific signal to stimulate the activation of Src at the GC, resulting in the phosphorylation of Aurora-A at Tyr148, a residue that is distinct from the canonical Thr288 activation site. The Src-mediated phosphorylation stimulates the kinase activity and also the recruitment at the CEs (14). Thus, we tested the hypothesis that forcing the cells to enter mitosis with an intact ribbon could result in reduced Aurora-A recruitment, which is necessary for the CE-based functions of this kinase (16). To this purpose, we used HeLa cells as we have previously addressed the role of Aurora-A and the experimental conditions to precisely control its expression in this cell model (14, 38, 44). The cells were synchronized for G2/M transition and treated with SP and MK in the absence or presence of BFA. The cells were fixed and stained for immunofluorescence with antibodies against pericentrin and Aurora-A (Fig 6A) to measure Aurora-A fluorescence levels at the CEs during G2 and prophase (38). Aurora-A activity can be stimulated by multiple mechanisms, including phosphorylation of sites distinct from Thr288 and binding to allosteric regulators, which operate only after its recruitment (52). Therefore, the level of recruitment at the CEs is a reliable proxy of Aurora-A activity (38, 52). Quantification of Aurora-A fluorescence intensity at the CEs during G2 and prophase showed that the treatment with SP and MK caused about a 40% reduction of its levels (Fig 6B). As a control, the fluorescence intensity

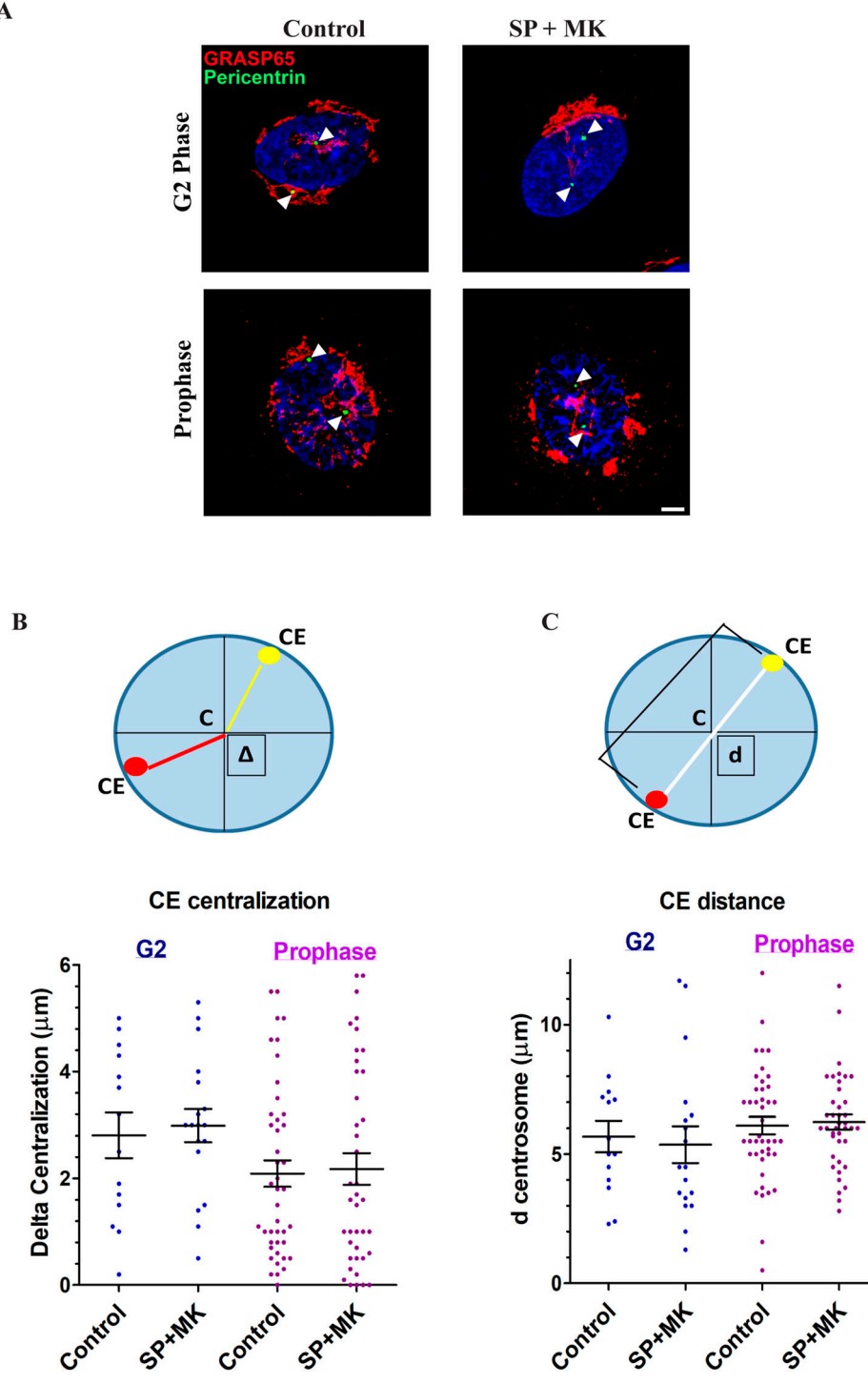

**A**

**Figure 4. Block of Golgi unlinking during G2 does not interfere with centrosomes separation and positioning.**
**(A)** HeLa cells were treated as described in Fig 1A and processed for immunofluorescence. Centrosomes localization was monitored in G2 and prophase by labeling the pericentriolar matrix with an anti-pericentrin antibody (green), GRASP65 for Golgi (red), and Hoechst 33342 to label the nuclei. Scale bar, 2 μm. **(B)** Upper panel: Schematic representation of the Δ parameter (delta centralization), evaluated as the differential distance of the two centrosomes from the center of the nucleus. Lower panel: Quantification of delta centralization in cells treated or not treated with SP and MK in G2 (blue) and prophase (purple); scatter plot distribution with average value ± SEM. The t test showed no significant differences between control and SP + MK–treated cells in both G2 and prophase. **(C)** Upper panel; schematic representation of the centrosomes separation, evaluated as the distance between the two centrosomes (d = distance). Lower panel: quantification of centrosomes separation in cells treated or not treated with SP and MK in G2 (blue) and prophase (purple); scatter plot distribution with average value ± SEM. t test showed no significant differences between Control and SP + MK-treated cells in both G2 and prophase.

of pericentrin during G2 and prophase did not change after treatment with SP and MK (Fig S5), indicating that the reduced Aurora-A recruitment was not an indirect effect of reduced availability of PMC. Aurora-A functional activity is not directly proportional to its local concentration but requires the reaching of a specific threshold to be effective (53), indicating that the observed reduction is physiologically relevant (53). Importantly, the reduction of Aurora-A recruitment

was a direct consequence of the lack of Golgi unlinking as it was fully recovered by BFA-induced GC disassembly (Fig 6B).

To examine whether the reduced level of Aurora-A at the CEs is a major cause of the spindle defects, we restored its local physiological concentration by transfecting Aurora-A-GFP. The precise control of Aurora-A expression is crucial because a high expression level can alter many mitosis-related events (16). Therefore, the experimental

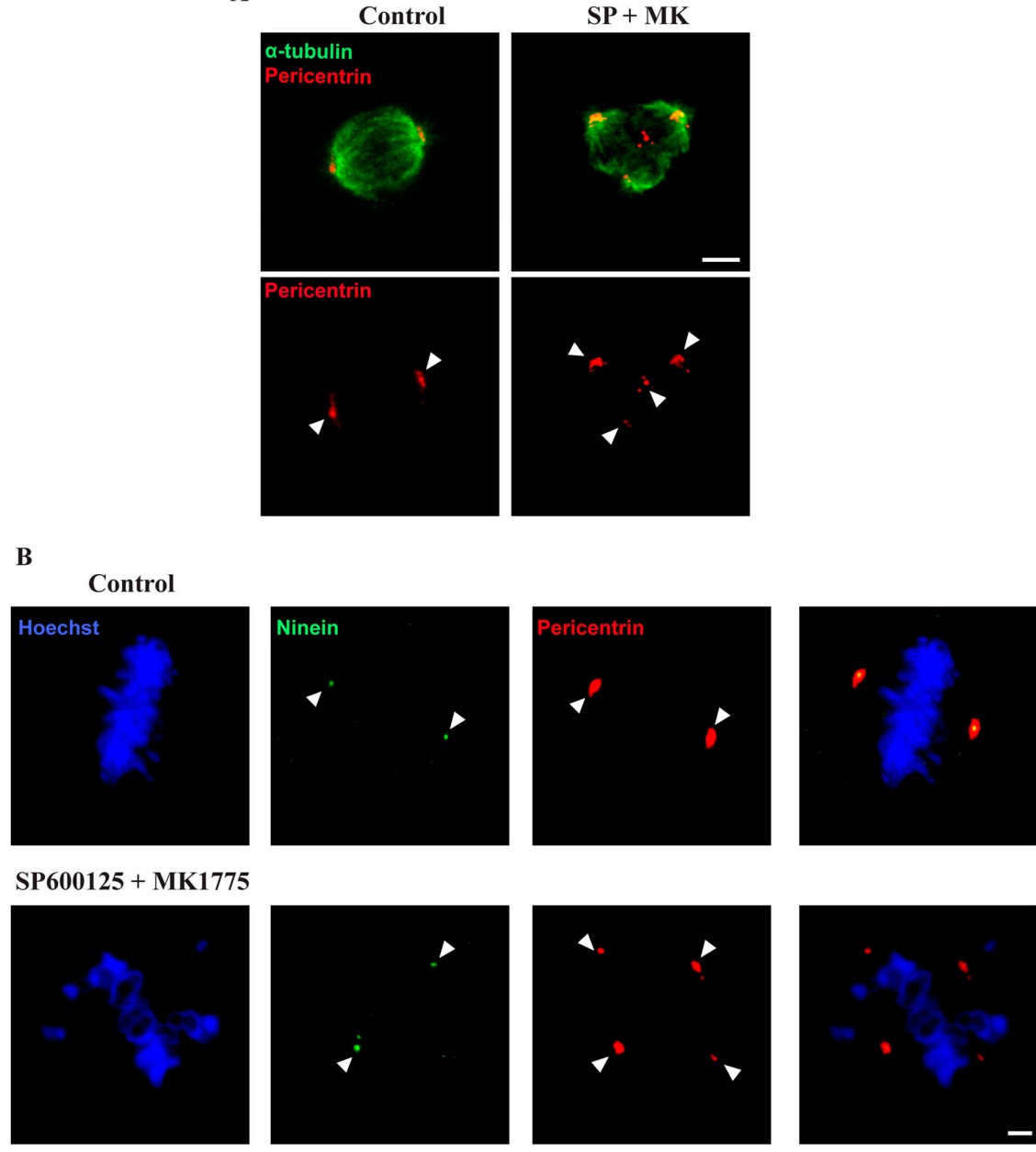

**Figure 5. Block of Golgi unlinking during G2 causes pericentriolar matrix (PCM) fragmentation.**
HeLa cells were treated as described in Fig 1A and processed for immunofluorescence. **(A)** The cells were labeled with antibodies against pericentrin and $\alpha$-tubulin to label the PCM and the microtubules, respectively. The cells treated with SP and MK showed an increased number of PCM-based MT nucleation foci compared with the control cells. Scale bar, 5 $\mu$m. **(B)** Cells were labeled with antibodies against pericentrin and Ninein to label the PCM and the centrosomes, respectively, and with Hoechst 33342 to label the DNA. Compared with the control cells, the cells treated with SP and MK showed an increased number of PCM-based MT nucleation foci but not of centrosomes. Scale bar, 2 $\mu$m.

conditions were fine-tuned to induce expression levels of exogenous Aurora-A similar to the endogenous protein, as previously described (38). As shown in Fig 6C and D, treatment of HeLa cells with SP and MK induced aberrant spindle in about 74% of the cells, in comparison to about 25% in untreated cells. Importantly, transfection of Aurora-A in HeLa cells synchronized for mitotic entry, induced a substantial recovery, about 50%, of correct spindle formation. These results indicate that a major cause of the GC-dependent spindle alterations

is an insufficient level of Aurora-A activation at the CEs during G2/prophase, likely resulting in defective control of MT polymerization or stabilization the at the PCM (51, 54).

**Golgi unlinking is also required for cytokinesis**

Defective spindles can result in cell death or abnormalities during mitotic exit (55). Therefore, we also examined the effects on

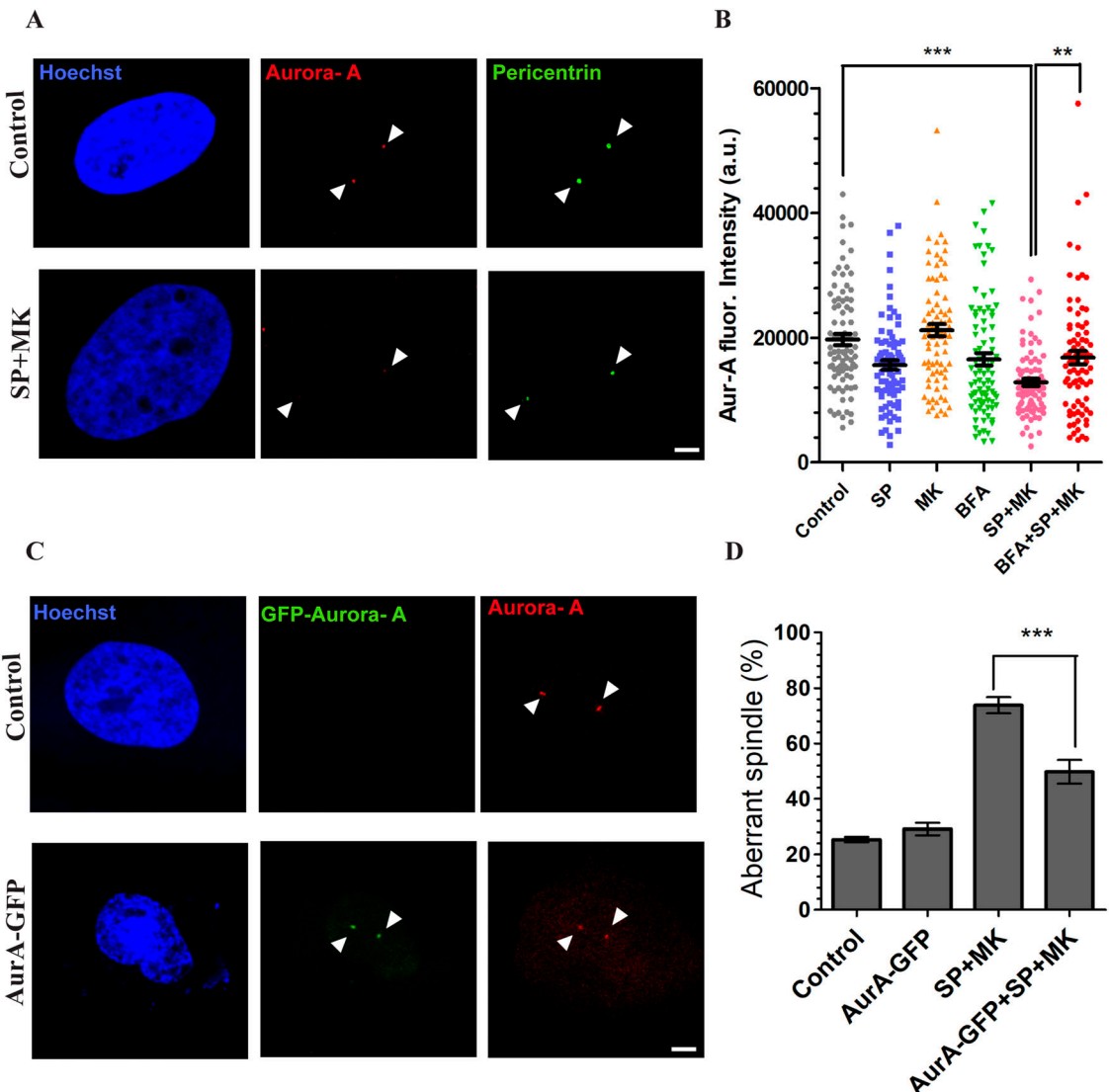

**Figure 6. Golgi unlinking controls spindle formation through Aurora-A.**
**(A)** HeLa cells were treated as described in Fig 1A and processed for immunofluorescence. Representative images of cells treated with vehicle or with the combination of SP and MK. The cells were fixed and processed for immunofluorescence with antibodies against Aurora-A (red) and pericentrin (green) to label the centrosomes. Scale bar, 2 μm. **(B)** Quantification of fluorescence intensity of Aurora-A (Aur-A) in cells treated as described. All the images analyzed were acquired at maximal resolution under fixed-imaging conditions. Equal areas were used to select the centrosome regions. Quantifications are represented as scatter plots and mean values ± SEM from three independent experiments. t tests were applied to the samples control versus SP + MK (***P < 0.0001) and SP + MK versus BFA + SP + MK (**P < 0.001). **(C)** HeLa cells were arrested in S-phase with double-thymidine block and transfected for 24 h with the empty vector or Aurora-A-GFP after the first release from thymidine. After fixation, the cells were labeled for immunofluorescence with antibodies against Aurora-A (red) and Hoechst 33342 to label the nuclei. **(D)** Quantification of the spindle defects (percentage of aberrant spindles) in cells treated with vehicle or with SP + MK and transfected with empty vector or Aurora-A-GFP. Scale bars: 5 μm. Data are means ± s.e.m. from three independent experiments. One-way ANOVA with Tukey's multiple comparison test. ***P < 0.0001 (SP + MK versus AurA-GFP + SP + MK).

postmitotic events after entering into mitosis with an intact GC in G2. To this end, NRK cells were synchronized for G2/M enrichment and treated to force entry into mitosis with an intact GC, as described in Fig 2A, with the difference that the cells were fixed 16 h after the mitotic peak as this time is sufficient to allow the duplication of most of the cells. The fixed cells were labeled for immunofluorescence with antibodies against GRASP65 and α-tubulin to detect the GC and the MTs, respectively, and Hoechst to visualize the DNA. Strikingly, confocal imaging showed that after forcing mitotic entry with an intact GC (SP + MK), more than 60% of the cells

became binucleated (Fig 7A and B). MK alone did not induce significant alterations of cell duplication. At the same time, the treatment with SP alone resulted in about 25% of binucleated cells (Fig 7B), in line with previous observations (56). Importantly, BFA-induced GC disassembly rescued cell division to levels similar to SP-treated cells, indicating that the block of Golgi unlinking is the direct cause of more than 50% of the observed binucleation events (Fig 7B).

We also tested this conclusion with an independent approach. RPE-1 cells were depleted of GRASP55 to induce irreversible Golgi

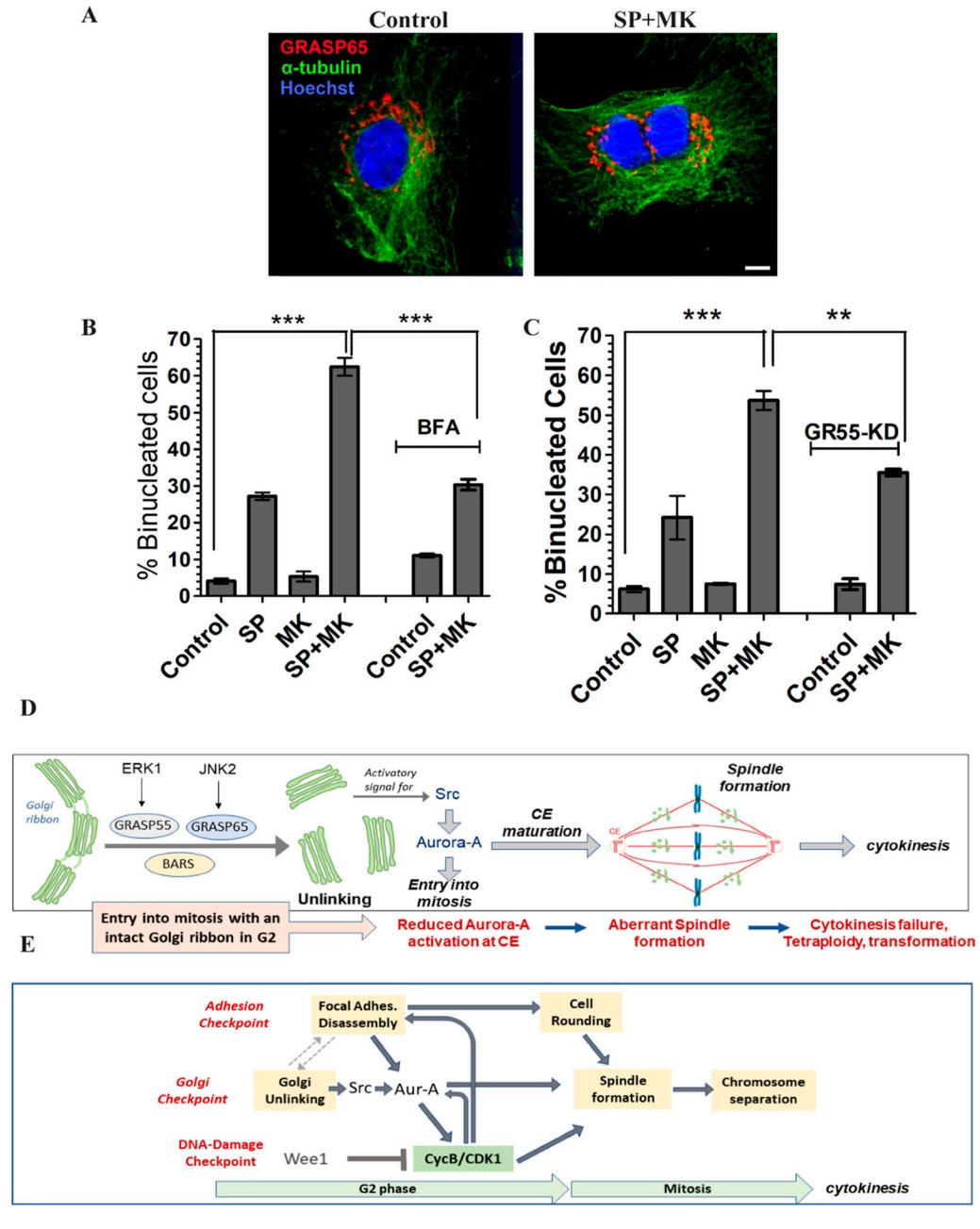

**Figure 7. The block of Golgi unlinking causes binucleation and transformation.**
**(A, B)** Normal rat kidney cells were treated as described in Fig 1A and fixed 24 h after aphidicolin washout. **(A)** Representative confocal images of cells treated with vehicle or the combination of SP and MK. The cells were fixed and labeled with antibodies against GRASP65 and α-tubulin to reveal the GC and microtubules, respectively, and with Hoechst to observe the nuclei. Scale bar, 10 μm. **(B)** Quantification of the fraction (%) of binucleated cells treated as described in (A). Data are means ± s.e.m. from three independent experiments. ***$P < 0.0001$ (control versus SP + MK). ***$P < 0.0005$ (SP + MK versus BFA + SP + MK). **(C)** RPE-1 cells were transfected with control siRNA (non-targeting, control; left panel) or with GRASP55-specific siRNA (GR55-KD; right panel) and treated as described in (Fig 3A). The fraction (%) of binucleated cells was quantified. Data are means ± s.e.m. from three independent experiments. $t$ tests were applied: ***$P < 0.0001$ (control versus SP + MK) and **$P < 0.005$ (SP + MK versus GR55-KD + SP + MK). **(D)** Schematic representation of Golgi unlinking and functional consequences on centrosome maturation, spindle formation, and cytokinesis. See the text for details. **(E)** Golgi and the de-adhesion checkpoints coordinate focal adhesion disassembly and Golgi unlinking to control cell rounding and spindle formation.

unlinking (27, 57) and synchronized for entry into mitosis by the double-thymidine block. The cells were forced to enter mitosis with an intact GC, as previously described, and fixed 16 h after the mitotic peak. The cells were stained with antibodies against GRASP55 and α-tubulin to label the GC and the MTs, respectively, and Hoechst to

label the DNA. MK alone did not alter the binucleation index, whereas SP caused about 22% of binucleation. The combined treatment with SP and MK resulted in more than 55% of binucleated cells (Fig 7C). Importantly, in cells with constitutively unlinked GC (GR55-KD), the normal duplication was rescued to levels analogous

to the treatment with SP alone (Fig 7C), further confirming that a significant fraction of the binucleation events is the direct result of the lack of Golgi unlinking.

On the other hand, binucleation can be the consequence of either failure of midbody cleavage or endomitosis, the latter consisting of the formation of two nuclei without cell division (37, 58). Thus, to distinguish between these two possibilities, the cells were synchronized, treated to force entry into mitosis in the absence or presence of BFA, and fixed 90 min after the mitotic peak. This is a suitable timing for measuring metaphase and cytokinesis events, the last-mentioned defined by the formation of the cleavage furrow. After fixation, the cells were stained with α-tubulin to visualize the cleavage furrow, GRASP65 to observe the GC, and Hoechst to label the DNA (Fig S6A). The quantification of the results showed that, compared with control cells, BFA did not alter the fraction of cells in metaphase or cytokinesis (Fig S6B). Treatment with SP reduced both populations, in line with the inhibitory effect on mitotic entry. At the same time, MK increased the fraction of cells in cytokinesis of about two times, a likely consequence of the stimulation of mitotic entry. Finally, the treatment with the combination of SP and MK induced an increase in the number of cells in metaphase, about the double compared with control cells, indicating a delay of the metaphase/anaphase transition, with a consequential and partial reduction of cells in cytokinesis. Importantly, the addition of BFA to cells treated with SP and MK, a treatment that rescues normal cell division, did not change the numbers of cells in cytokinesis compared with cells treated with SP and MK alone (Fig S6B). Therefore, the BFA-induced rescue of normal cell division in cells treated with SP and MK does not correlate with an increased number of cytokinesis events, indicating that endomitosis is not the mediator of the Golgi-dependent binucleation, which likely could result from failed midbody cleavage. In addition, the BFA-mediated recovery of correct cell duplication does not cause cell death, as evidenced by the counting at the microscope of the number of nuclei per field of view (n/f.o.v.) 24 h after aphidicolin washout (Fig S6C), indicating that BFA does not cause an apparent recovery of cell duplication by death-mediated elimination of the failed mitotic events.

Altogether, these data indicate that the substantial increase of binucleation observed in cells treated with SP and MK could be the direct consequence of midbody cleavage failure in cells with an intact GC ribbon in G2. Binucleation can have critical pathological consequences as the associated tetraploidization, even if transient, can lead to chromosomal instability, aneuploidy, and tumorigenesis (59).

## Discussion

In this study, we revealed the physiological role of GC unlinking in G2 showing that it is necessary for the fidelity of later mitotic events, including correct bipolar spindle formation and cytokinesis (Fig 7D). Indeed, forcing entry into mitosis of cells with an intact Golgi ribbon in G2 resulted in spindle multipolarity and tetraploidization. These defects were not the indirect effect of drag forces exerted by a bulky unlinked GC but the direct consequence of the lack of Golgi unlinking in G2 and of the consequently reduced activation of a Golgi-based signaling that leads to recruitment and activation of the kinase Aurora-A at the CEs. Finally, our findings revealed that alterations of GC segregation in G2 could have critical pathological consequences as they can induce cell transformation.

To investigate the role of Golgi unlinking in cell division, we have developed a simple strategy based on the stepwise addition of drugs and a series of controls to identify the physiological consequences of entry into mitosis with an intact GC. In brief, first, the JNK inhibitor SP600125 (SP) is added to cells enriched in G2 to block Golgi unlinking and induce G2 arrest (29). Then, the Wee1 inhibitor MK1775 (MK) is added to activate CDK1 and trigger a rapid entry into mitosis (35). The combined treatment with SP and MK caused severe spindle multipolarity and binucleation, being both the direct result of the block of Golgi unlinking as they are rescued by satisfying the Golgi checkpoint through BFA-mediated GC disassembly (29) or constitutive unlinking after GRASP55 depletion (57). Thus, our results indicate that the specific structural organization of the GC in G2 (i.e., intact ribbon or disassembled stacks) determines the correct accomplishment of subsequent mitotic events, including spindle formation and the completion of cell abscission.

Our experimental procedure has been set up to pinpoint the role of the G2-specific GC unlinking on mitosis. Thus, it is different from the approach developed by Seemann's group, which was based on blocking stacks disassembly after the mitosis onset through the induction of large polymers formation in the CG lumen (23). Supporting the difference between the two approaches, the Golgi membranes, even if in the presence of SP, are extensively disassembled after the forced entry into mitosis with MK, thus excluding the possibility that the presence of an intact "bulky" Golgi ribbon in G2 could alter spindle formation because of a drag force exerted on CE positioning during G2/prophase. Golgi disassembly in the presence of SP is likely caused by the activation of CDK1, which stimulates Golgi vesiculation (18). In addition, our data show that the forced entry into mitosis with an intact GC causes the formation of multipolar and disorganized spindles. Conversely, the block of stacks disassembly during mitosis, as shown by Seeman's group, causes the formation of monopolar spindles, spindle assembly checkpoint activation, and metaphase arrest (23).

An important question from our data is how Golgi unlinking in G2 affects spindle formation and cytokinesis. The spindle comprises three types of fibers, known as polar, astral, and kinetochore MTs, which have specialized tasks and are oriented in different directions. Many complex signals coordinate the lengthening and shortening of the various MT fibers that, together with the action of minus- and plus-directed MT motors, concur to build a bipolar spindle with the chromosomes aligned at the spindle midzone (60).

Despite the complexity of spindle formation, our data offer a mechanistic explanation for the formation of the aberrant spindles. Indeed, investigating the role of GC unlinking in the regulation of mitotic entry, we have previously shown that ribbon unlinking acts as a signal to trigger Src activation. In turn, Src phosphorylates Tyr148 of Aurora-A, stimulating its recruitment at the CEs and kinase activity (14) through a mechanism distinct and independent of the phosphorylation of Thr288, which is the primary activation site (16). Aurora-A is essential for many distinct mitotic events, each of which is regulated by a dedicated mechanism of activation that involves the recruitment of this kinase at precise subcellular locations, where it becomes active and able to phosphorylate a definite set of

effectors (16, 61). In addition to stimulating entry into mitosis, Aurora-A is the main regulator of the formation of spindle fibers (51, 54). At the G2/prophase transition, Aurora-A becomes associated with the PCM, a large multimeric structure composed of scaffold proteins that organize the anchoring, polymerization, and stabilization of newly nucleated MTs (61).

Notably, we found that the cells forced to enter mitosis with an intact ribbon showed reduced levels of Aurora-A at the CEs, and this phenotype was substantially recovered by artificial GC disassembly. Besides, by restoring the CE levels of Aurora-A through the transient transfection of Aurora-A, we observed a prominent rescue of spindle formation. These results indicate that a major cause of spindle alterations is the insufficient level of Aurora-A activation at the CEs because of the lack of Golgi unlinking in G2 (51, 54). In addition, we found that defects in pre-mitotic Golgi segregation could have significant physiological effects as the mitotic spindle multipolarity, the consequent chromosomal instability, and the cytokinesis failure cause a transient tetraploidy, which can contribute to aneuploidy (59, 62). This latter condition promotes genomic instability, an important risk factor for cell transformation and tumorigenesis (59, 63).

In conclusion, our findings show that Golgi unlinking is a crucial preparatory step for mitotic entry and progression. An additional preparatory step for mitosis is the "adhesion checkpoint" (Fig 7E), which is involved in the control of cell reshaping from a flat to a spherical geometry (2). The G2-restricted expression of the scaffold protein DEPDC1B induces the selective disassembly of a subset of FAs (Fig 7E) (4), which is required for the retraction of the cell margins and the formation of a rigid actomyosin cortex that directs the process of cell rounding in metaphase (2). The latter process creates an optimal geometry for spindle orientation and assembly, which is essential for correct chromosome segregation (2). FA disassembly also allows the relocation to the CEs of scaffold proteins, such as NEDD9 and Ajuba, which protect the phosphorylation of Aurora-A in Thr288 by phosphatases (Fig 7E) (17).

These findings suggest that Aurora-A works as an integrator of different stimuli, which are originated at the FAs and the GC, to induce PCM assembly and to activate CycB-CDK1 for mitotic entry (16) (Fig 7E). Notably, the downstream target of Aurora-A, CycB-CDK1, participates to FA disassembly (64).

Overall, our data offer a novel model for the regulation of G2/M transition. The major regulator of the mitosis onset is the synthesis and accumulation of Cyclin B, which forms a complex with the kinase CDK1. The complex is kept inactive by the inhibitory phosphorylations of CDK1 operated by the kinases Myt-1 and Wee1, whereas the dephosphorylation is carried out by members of the CDC25 phosphatase family, which are stimulated by Aurora-A. The equilibrium between phosphorylation and dephosphorylation of the inhibitory residues of CDK1 is controlled by different physiological conditions. The activation of DNA replication/damage checkpoints induces the inhibition of CDK1 until DNA has been completely and correctly replicated (1). In addition, the proper reshaping is required for correct chromosome segregation, and this physiological need is controlled by the Golgi and adhesion checkpoints, whose satisfaction tunes the equilibrium toward CDK1 activation thanks to the Aurora-A–mediated stimulation of CDC25 (Fig 7E).

Thus, our data suggest that the Golgi and the adhesion checkpoints are part of a series of coordinated signaling circuits among

the GC itself, the FA and the CEs devoted to preparing the cells for the extensive cellular morphological reorganizations that are required to ensure proper cell rounding and correct spindle orientation and chromosome inheritance (Fig 7E), which is fundamental also for tissue development and maintenance (2). As an additional value, our findings can also open new avenues for cancer therapy as the identification of a novel mechanism involved in controlling spindle formation can be exploited to induce mitotic catastrophe in tumor cells (65) by defining a combinatorial chemotherapeutic strategy able to target Golgi unlinking and other crucial mitotic functions, increasing drug potency and efficacy and enhancing antitumoral activity.

As a final consideration, the GC structure varies across different species and tissues. In the yeast *Saccharomyces cerevisiae*, the GC is organized as individual and dispersed cisternae (66). In contrast, in the yeast *Pichia pastoris*, in plants, and drosophila, the GC is organized as stacked cisternae (67, 68). Interestingly, in Drosophila S2 cells, the Golgi stacks are duplicated as twins during S-phase. Separation of the stacks composing the twins is required for G2/M transition (69). Moreover, the structure of the GC can vary also between different cell types of the same organism. For example, in differentiated neurons, in addition to the perinuclear GC, satellite Golgi outposts in dendrites control dendritic morphogenesis and membrane traffic to synapses (70). Also, during muscle cell differentiation, the GC is organized in the form of scattered isolated stacks (71). Less investigated is the structure of the GC in differentiated epithelia. Therefore, it is possible to speculate that in differentiated and nondividing cells, the ribbon structure becomes superfluous and that this structure is an exclusive characteristic of cells in the active phase of division and not fully differentiated. Future studies in organoids or model organisms and the development of selective approaches to inhibit GC ribbon separation could help to clarify this hypothesis.

# Materials and Methods

### Cell cultures

NRK cells (Cat# 86032002, RRID:CVCL_3758; ECACC) were from the American Type Culture Collection (Manassas, Virginia) and HeLa (Cat# 93021013, RRID:CVCL_0030; ECACC) cells were from the European Collection of Authenticated Cell Cultures (Manassas, Virginia) and were cultured in DMEM and MEM, respectively (Invitrogen), supplemented with 10% FCS (Biochrom), 100 $\mu$M minimal essential medium nonessential amino-acid solution, 2 mM L-glutamine, 1 U/ml penicillin, and 50 $\mu$g/ml streptomycin (all Invitrogen). hTERT-RPE1 (kindly provided by B. Franco, TIGEM) were grown in DMEM/F12 with 10% FBS (Gibco, BRL), 2 mM l-glutamine, 1 U/ml penicillin, and 50 $\mu$g/ml streptomycin (Invitrogen). All the cell lines were grown under a controlled atmosphere in the presence of 5% $CO_2$ at 37°C.

### Synchronization protocol

HeLa and RPE-1 cells are synchronized using double-thymidine cell cycle arrest: the cells were maintained in growth medium plus 2 mM

thymidine for 16 h and then rinsed and maintained in growth medium for 8 h. The cells were then maintained in thymidine for additional 16 h before the final drug washout. For the cell cycle synchronization of NRK cells, the population was incubated with aphidicolin O/N, and the day after, cells were released from the aphidicolin block. The G2/M phase of NRK and HeLa cells was reached after 8 h from the washout.

## Mitotic index assay

For mitotic index analysis of synchronized cells, the cells were plated at 60% confluence on 15-mm coverslips. The mitotic index was estimated by measuring the number of cells, stained with Hoechst 33342 and showing clear mitotic (condensed chromosomes) and interphase (diffuse nuclear staining) features, using a confocal laser microscope (Zeiss LSM700 confocal microscope system; Carl Zeiss) equipped with ×63 1.4 NA oil objective.

## Flow cytometry

NRK cells were synchronized with aphidicolin and RPE-1 cells with the double-thymidine cell cycle arrest protocol. Trypsinized cells were pelleted, washed in cold PBS, and resuspended in ice-cold ethanol while vortexing. The cells were incubated O/N at 4°C. The next day, the ethanol was removed by centrifugation and the cells were washed in cold PBS and incubated with 50 µg/ml propidium iodide (Invitrogen) for 30 min in the presence of RNASe (Sigma-Aldrich). The cells were then analyzed using the Becton Dickinson FACSCantoA instrument (BD). The data were plotted with Diva software, with 20,000 events analyzed for each sample.

## Antibodies and reagents

Aphidicolin (2.5 µg/ml), thymidine (2 mM), brefeldin A (200 ng/ml), SP600125 (25 µM), and fibronectin (10 µg/ml) were from Sigma-Aldrich. MK1775 (0.3 µM) was from Selleckchem. DMSO was from Carlo Erba. Hoechst 33342 was from Invitrogen. Mowiol 4-88 was from Sigma-Aldrich. The antibodies were from the following sources: anti-GRASP65 (Cat# ab48533, RRID:AB_880296; Abcam) (1:3,000) and anti-pericentrin (Cat# ab4448, RRID:AB_304461; Abcam) were from Abcam; anti-α-tubulin (Clone DM1-A) (RRID:AB_3067992) (1:5,000) was from Sigma-Aldrich; anti-GRASP55 was from Novus Biologicals (Cat# NBP1-00862, RRID:AB_1503318; Novus) (1:500); anti-Aurora-A was from BD Transduction Laboratories (1:200); anti-Ninein was from BioLegend (1:200) (Cat# 602802, RRID:AB_2251471; BioLegend). Alexa 488– and Alexa 568–conjugated secondary antibodies (Cat# 710369, RRID:AB_2532697, and Cat# A-11004, RRID:AB_2534072, respectively) (1:400) were from Thermo Fisher Scientific.

## Plasmids, siRNAs, and transfection

The GRASP55 vector was targeted using a smart pool from Dharmacon directed against the sequences: #1 5′-GGAGUGAGCAUUCGUUUCU-3′; #2,5′-GUAAACCAGUCCCUCACUU-3′; #3,5′-GACCACACAGUGAUUAUAU-3′; #4,5′-UGUCGAGAAGUGAUUAUUA-3′. The siRNA duplexes were transfected using Lipofectamine 2000 (Invitrogen), according to

the manufacturer instructions. As a non-targeting (control), we used the siRNA duplex 5′-UUCUCCGAACGUGUCACGUdTdT-3′ (Sigma-Aldrich). The cells were further processed according to the experimental design, and depletion was assessed by immunofluorescence. For transfection of Aurora-A-GFP, HeLa cells were arrested in S-phase with double-thymidine block and transfected in 24 well with 0.25 µg of vector for 24 h with the TransIT-LT1 Transfection Reagent (Mirus), according to the manufacturer's instructions. The cDNA of full-length Aurora-A fused to GFP was a kind gift from Dr. Stefano Ferrari (Institute of Molecular Cancer Research, University of Zurich, Zurich, Switzerland).

## Immunofluorescence microscopy

The cells were grown on 10 µg/ml fibronectin-coated glass coverslips (Sigma-Aldrich) and treated as described above. They were then fixed with 4% PFA (Electron Microscopy Sciences, Hatfield, Pennsylvania) for 10 min at RT. The blocking buffer (0.5% BSA, 0.1% saponin, 50 mM NH$_4$Cl in PBS supplemented with sodium azide) was then added to the cells for 20 min. For the labeling of α-tubulin, the cells were fixed with methanol at –20°C for 7 min and blocked with 1% BSA in PBS. In any case, the samples were washed in PBS and incubated for 1 h at RT or overnight at 4°C with the primary antibodies in the blocking buffer. The secondary antibodies plus 2 µg/ml Hoechst were incubated for 40 min at RT. The coverslips were then mounted on glass microscope slides with Mowiol 4-88 (Sigma-Aldrich). The cells were fixed and stained with Hoechst 33342 for labeling of the nuclei. Immunofluorescence samples were examined using a confocal laser microscope (Zeiss LSM700 confocal microscope system; Carl Zeiss) equipped with ×63 1.4 NA oil objective. Optical confocal sections were taken at 1 Airy unit, with a resolution of 512 × 512 pixels or 1,024 × 1,024 pixels, and exported as .JPEG or .TIF files. The images were cropped with Adobe Photoshop CS6 and composed using Adobe Illustrator CS6. High-resolution images were acquired at the Zeiss LSM880 using an Airyscan detector and further processed with deconvolution algorithms of Zeiss Zen Black 2.1 software.

## Quantitative analysis of GC area fragmentation

The analysis was performed using ImageJ. Images were subjected to thresholding, and the number of particles ("Golgi objects") was calculated with the Analyze Particles function. All the images were acquired at maximal resolution, under fixed-imaging conditions. For qualitative analysis of the phenotype of the GC, the GRASP65 staining was processed with the freeware ImageJ software package using the "Invert" function.

### Centrosomes distance parameters

Centrosome position was determined using ImageJ. Both centrosomes were tracked in G2 and prophase. Centrosome-to-centrosome distance (CE distance) was calculated by measuring the distance of centrosome foci in a single cell.

### Centrosomes delta centralization

The differential distance from the center of nucleus of the two centrosomes was measured in G2 and prophase, using ImageJ. The

center of nucleus is identified as equidistal distance from the cell edge. The distance of centrosomes from the nucleus is measured, calculating the distance of each centrosome from the center of the nucleus.

### Fluorescence intensity of Aurora-A

The analysis was performed using ImageJ. Cells were imaged with a confocal laser microscope (LSM710, Carl Zeiss; objective: 63 × 1.4 NA oil; definition: 1,024 × 1,024 pixels). The bright centrosomal regions identified by a centrosome marker (pericentrin) were circled, the Aurora-A fluorescence intensity in these regions and in a similarly sized background region were determined using LSM710 software (ZEN 2008 SP1), and the Aurora-A centrosomal fluorescence was calculated from these values.

### Statistical analysis

Statistical significance was determined by an unpaired, two-tailed, $t$ test.

## Supplementary Information

## Acknowledgements

The authors would like to thank the Italian Association for Cancer Research (AIRC, Milan, Italy; IG 2017 id. 20095 to A Colanzi), the POR FESR Campania SATIN and Next Generation EU, in the context of the National Recovery and Resilience Plan, Investment PE8—Project Age-It: "Ageing Well in an Ageing Society" (CUP: B83C22004880006) for financial support, and the BioImaging Facility at the Institute of Experimental Endocrinology and Oncology for support in imaging microscopy.

### Author Contributions

F Mascanzoni: conceptualization, data curation, investigation, and writing—original draft.
I Ayala: conceptualization and writing—review and editing.
R Iannitti: formal analysis, validation, and writing—review and editing.
A Luini: conceptualization and writing—review and editing.
A Colanzi: conceptualization, supervision, funding acquisition, and writing—original draft and project administration.

### Conflict of Interest Statement

The authors declare that they have no conflict of interest.

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
