## [Reviewer comments · Life Science Alliance]

Life Science Alliance

The Golgi checkpoint: Golgi unlinking during G2 is necessary for spindle formation and cytokinesis

Fabiola Mascanzoni, Inmaculada Ayala, Roberta Iannitti, Alberto Luini, and Antonino Colanzi

DOI: <https://doi.org/10.26508/lsa.202302469>

Corresponding author(s): Antonino Colanzi, Institute of Experimental Endocrinology and Oncology "G. Salvatore" (IEOS) and Alberto Luini, Institute for Experimental Endocrinology and Oncology

Review Timeline:

Submission Date:	2023-11-03
Editorial Decision:	2023-12-04
Revision Received:	2024-02-20
Editorial Decision:	2024-02-22
Revision Received:	2024-02-28
Accepted:	2024-02-28

Transaction Report:

December 4, 2023

Re: Life Science Alliance manuscript #LSA-2023-02469-T

Dr. Antonino Colanzi
Institute of Experimental Endocrinology and Oncology "G. Salvatore" (IEOS)
National Research Council
Via Pietro Castellino 111
Naples 80131
Italy

Dear Dr. Colanzi,

Thank you for submitting your manuscript entitled "The Golgi checkpoint: Golgi unlinking during G2 is necessary for spindle formation and cytokinesis" to Life Science Alliance. The manuscript was assessed by expert reviewers, whose comments are appended to this letter. We invite you to submit a revised manuscript addressing the Reviewer comments.

Thank you for this interesting contribution to Life Science Alliance. We are looking forward to receiving your revised manuscript.

Sincerely,

B. MANUSCRIPT ORGANIZATION AND FORMATTING:

Reviewer #1 (Comments to the Authors (Required)):

The organization of Golgi membranes into stacks of cisternae and the stacks into a ribbon in the pericentriolar position is fascinating. What is the role of this organization and why should anyone care? Decades have been spent understanding the mechanism and the physiological significance of cisternal stacking and there is little in the way of a concrete answer. These membranes undergo extensive fragmentation at the onset of mitosis and for decades it has been proposed that this is necessary for partitioning of the membranes during cell division. But is it? A clear experiment to test this proposal is lacking. A newish proposal is that these membranes fragment to ensure spindle dynamics. There is a good reason to believe in this data because Golgi membranes have the capacity to nucleate microtubule esters, which could potentially cause havoc for a cell to build a bipolar spindle in metaphase. May be then Golgi fragmentation is way to prevent the interference in assembly of a bipolar spindle.

I like this work because it provides insights into the role of Golgi in spindle dynamics in association with mitotic events.

I don't have any issues with the technical aspects of the paper. There is a heavy use of pharmacology, but this is likely the best approach to prevent cells from adapting to alternatives in molecular approaches of knockdown or over expression of genes.

1. What is CE? The authors throw this abbreviation without explaining anywhere what it defines.
2. In cells that contain an intact Golgi and allowed to enter mitosis by affecting the activity of CDK1, would the cells enter mitosis upon treatment with a small dose of nocodazole? Nocodazole depolymerises microtubules and causes stacks of Golgi to separate and disperse in the cytoplasm. Does short term treatment with nocodazole activate Src?
3. The authors are encouraged to include a paragraph on why this system is important for a subset of cells. Yeast and fully polarised and terminally differentiated cells do not organise their Golgi in the form seen in fibroblasts and hence might not need this mechanistic control.
4. Is there any way to know what comes first: dissolution of the focal adhesions or the separation of the Golgi ribbon into stacks?

Otherwise, this is a fine piece of work that should be published.

Reviewer #2 (Comments to the Authors (Required)):

In this manuscript, the authors developed a novel strategy to interrogate the requirement for Golgi ribbon separation as a mitotic checkpoint. This strategy involves cell cycle synchronization, inhibition of JNK2 with SP600125 to prevent the Golgi unstacking in G2, and then activation of CDK1 using the Wee1 inhibitor MK1775. This new method is likely to be useful to others in the field. Here, the authors use this strategy to provide more evidence for the importance of Golgi unstacking as a mitotic checkpoint. Importantly, the authors show that when mitosis is induced with an intact Golgi there is an increase in mitotic defects, particularly spindle multipolarity and defective cytokinesis resulting in binucleation. Additional experiments suggested a mechanism for these defects is lack of recruitment of Aurora-A kinase at centrosomes.

The experiments are mostly convincing and well-controlled. A few points should be addressed by the authors:

1. The final piece of data (Figure 6D) regarding increased cellular transformation is potentially very interesting but not very compelling in the present form. It appears that NIH3T3 cells were treated one time with SP and/or MK and/or BFA and then they were washed out and cells were allowed to grow for 2 weeks to form foci. It is a bit difficult to understand how this one time treatment would lead to increased foci formation over the course of 2 weeks. The authors may need to explain the assay and interpretation better. In addition, the foci numbers went from 10 in the control to 20 in the SP+MK condition - the authors should show images of the crystal violet staining of representative plates so the reader can better visualize the assay.
2. The authors use different cells in different figures, including NRK, HeLa and TERT-RPE1. The authors should provide some

rationale as to why these different cell lines are being used in different assays.

3. Please describe in the methods section how the mitotic index assay is performed.

4. Please check and define all abbreviations. It was not clear that some abbreviations, such as CE and SAC, were defined for the reader.

Reviewer #3 (Comments to the Authors (Required)):

In this paper, Mascanzoni et al. investigate a possible implication of Golgi ribbon cleavage in mitotic spindle formation and cytokinesis. Previous work from the Colanzi's group reported that the TGN-associated kinase Src is activated at G2/M by Golgi unlinking, and that this activated Src phosphorylates mitotic Aurora-A kinase, thus promoting its recruitment to the centrosome and its activity. Subsequent CDK1 activation triggers mitotic entry. They also described that JNK2 has a crucial role in Golgi cleavage during G2 through phosphorylation of the Golgi protein GRASP65 on Ser 277. Based on these results, they now combine JNK2 inhibition (to block Golgi unlinking) with CDK1 activation (to stimulate mitosis entry) and analyse subsequent perturbations in cell division. They conclude that Golgi cleavage-induced Aurora-A signalling is crucial for correct spindle formation and cytokinesis.

This work tackles a very interesting question: how mitotic events at different subcellular structures are regulated and coordinated in order to achieve a successful cell division. The authors conclude that Golgi ribbon cleavage acts a crucial player, not only at the G2/M checkpoint as previously reported, but it is also pivotal for proper spindle formation and faithful cell division and cytokinesis.

My major concern on the manuscript is the extensive use of kinase inhibitors as the main (almost the only) strategy to support conclusions. Both inhibition of JNK2 and stimulation of CDK1 might produce unrelated effect that might conceivably affect the interpretation of the results. To rule out this possibility, the authors employ BFA treatment. However, in my opinion the mere employment of BFA, although suitable, is not sufficient per se to demonstrate that the observed phenotypes are exclusively due to Golgi unlinking. In this regard, some Golgi proteins, such as golgin160, the receptor of the dynein/dynactin complex at the GA, strongly associate with the centrosome in the presence of BFA (Yadav et al., 2012). It is reasonable to believe that relocation of Golgi proteins after BFA treatment might alter centrosome activity. In fact, Figure 2C and 4B of this work show that BFA alone reduces Aurora A intensity at the centrosome and produces aberrant spindles, respectively. The authors reported in their previous paper Cervigni et al. 2015 that a single point mutation in GRASP65 (Ser277 phospho-mutant) recapitulates the effects on the Golgi structure observed after JNK2 inhibition. Therefore, the most relevant data of this work, notably those concerning to the centrosome and spindle poles, should be confirmed by using a similar approach. Furthermore, in order to strengthen the conclusions drawn, appropriate control should be included in most of the experiments performed as detailed below.

Major points:

o Why different cell lines are employed throughout the paper. NRK and HeLa are transformed cells, whereas RPE1 are primary immortalised cells. Both Golgi and centrosome composition and activity greatly vary among cell lines. For instance, the Golgi of RPE1 cells actively nucleates microtubules, while in HeLa cells this activity is highly variable or absent. Furthermore, the authors employ different synchronisation protocols depending on the cell type used. Since most of the experiments performed rely on an accurate G2 arrest and subsequent mitotic entry, it would be convenient to include a simple flow cytometry analysis resulting from these synchronisation methods. Detailed synchronisation protocols should also be indicated in the Materials and Methods section. Text and Figure 1 include different timing of drugs addition and cell processing. Text and Figure 1 refer to different timings of drug addition and cell processing.

o Figure 1.- The authors claim that the Golgi ribbon remains intact in the presence of kinase inhibitor SP600125 and in the presence of both SP600125 and MK1775. However, I would argue that a number of Golgi objects can be appreciated in both the top and bottom panels of Figure 1B. It would be interesting to quantify the number and area of the Golgi elements present in all these conditions and compare it with those of Golgi from G1 cells. This would provide a robust assay to evaluate the efficiency of the treatments employed, and show that the variability observed in Golgi elements is not greater than what normally observed in intact Golgi in NRK cells. Indeed, one would expect that values of G1 and G2-treated cells would be similar and lower than those of control G2 cells. In addition, it would be good to indicate how G2 cells were identified in each condition.

o Figure 2. Please explain what is the difference between cells showed in Figure 1B SP+MK and those showed in Figure 2D SP+MK. In the first case, the authors conclude that under these conditions the Golgi ribbon remained intact, while in the second one they state that the GA is highly fragmented probably due to CDK1 stimulation of Golgi disassembly pathways distinct from the JNK2-mediated GRASP65 phosphorylation. Accordingly, BFA has no effect on the area of Golgi fragments induced by SP+MK treatment. This issue has to be clarified.

Figure 2C. Representation of data is tricky and impairs the interpretation of the relevance of aberrant mitotic spindle in the different conditions. For instance, the number of aberrant spindles is higher in SP treated cells (that are blocked in G2) or in BFA treated cells (that enter mitosis with a disassembled GA) than in control cells. In order to appreciate the relevance of this experiment, I would suggest to report the percentage of aberrant spindles to the total spindles number showed in Figure 1C.

Also, statistical significance of control versus control BFA treated cells and control BFA versus BFA+SP+MK should be calculated and plotted.

o Figure 3. The experiments included in this figure are highly relevant to support the conclusions drawn from this work because it makes use of a different approach and employs a non-transformed cell line. Therefore, it is important to well characterize the behaviour of the GC after drug treatments in parallel with the number of aberrant spindles. The degree of Golgi fragmentation in control RPE1 and after GRASP55 KD under all the conditions tested has to be shown and quantified as before. Furthermore, cell synchronisation efficiency should be monitored through FACS analysis and reported. In my experience, double thymidine block is not the best method to synchronise RPE1 cells.

Does SP treatment on its own lead to aberrant spindle formation in GRASP55 KD cells? How do the authors explain the high number of aberrant spindles in SP+MK treated GRASP55 KD cells?

Figure 3A simply shows the efficiency of GRASP55 silencing and it is not directly relevant to the main conclusions of the manuscript. Therefore, it is highly recommended to move it to the supplemental material.

Figure 3B.: Same comment as for Figure 2C regarding data representation.

o Figure 4. Please show Golgi labelling in Figure 4A to estimate the degree of Golgi unlinking under these conditions. The dispersion of data points in Fig 4B and 4C is very high. How were G2 and prophase cells chosen for this analysis? Since PCNT is affected by SP+MK treatment (Figures 4 and 5), a centriolar marker as ninein may be more appropriate for this analysis.

o Figure 5. In this figure, another very relevant result sustaining the conclusions of the work is shown. SP+MK treatment reduces accumulation of Aurora A at the centrosome resulting in a significant increase in aberrant spindle percentage (from 20% to 80%). This effect was partially reverted by the ectopic expression of Aurora A. Therefore, the authors conclude that "a major cause of the GC-dependent spindle alterations is an insufficient level of Aurora-A activation at the CEs during G2/prophase...." However, SP+MK treatment seems to inhibit centrosome maturation, as can be appreciated through PCNT labelling of cells treated with SP+MK (shown in both figures 4A and 5A). Critical functions of Aurora A-kinase at the mitotic centrosomes are essentially mediated by the interaction with its cofactor Cep192. Since Cep192 accumulates at the PCM during centrosome maturation in G2, and this process is blocked by SP+MK treatment, it is difficult to distinguish whether the decrease in Aurora A recruitment is due to lack of Golgi unlinking-induced signalling or to inhibition of PCM maturation.

Given the complexity of the mechanisms underlying recruitment and activation of Aurora A at the centrosome, data obtained should be further confirmed by depleting GRASP55 or GRASP65 or, even better, by expressing a constitutively inactive version of GRASP65 (Ser277 phospho-mutant).

In my opinion, Figure S2 showing PCM fragmentation after SP+MK treatment and causing multipolar spindles should be included in the main text.

Figure 6. From my point of view, results shown in this figure are too preliminary and difficult to integrate with the rest of data. A deeper characterization is required to propose that Golgi unlinking has a role in midbody cleavage, especially taking into account that at this stage of mitosis, the Golgi ribbon has already reformed in control cells. In the same way, there are not enough data to support that "alterations of the spindle and the tetraploidization caused by the lack of GC disassembly in G2 can potentially induce cell transformation".

Figure 6A shows that SP treated cells are able to progress into mitosis. In that case, what is the advantage of adding MK inhibitor? Why aren't mitotic indexes and the rest of phenotypes analysed at a longer SP treatment timepoint? Actually, 1 hour after adding MK inhibitor, only 20% of control cells were in metaphase.

The authors should propose an explanation about the high number of binucleate cells in SP+MK treated GRASP55 KD cells knowing that MK does not affect mitotic exit. Does SP alone trigger binucleation in KD cells? How does this compare to control cells treated with SP alone? Significance of data SP versus SP+MK in KD cells should be included.

Minor points:

- In the first paragraph of the introduction, the authors must state what CE stands for, or alternatively, use centrosome throughout the paper.
- The graphs of all figures expressing percentages should have the same y-axis, possibly set to 100%.
- Numbers referring to the quantification provided in the graphs (i.e mitotic index observed in control cells (20% {plus minus} 5%) ...) should be included in the text.

Antonino Colanzi

Principal Investigator
Coordinator of the Second Unit of the
Experimental Endocrinology and Oncology Inst (IEOS);
National Research Council (CNR);
Via P. Castellino 111, 80131 Naples, Italy
Tel: ++39- 081.6132.538;
e-mail: a.colanzi@ieos.cnr.it

Reviewer #1):

The organization of Golgi membranes into stacks of cisternae and the stacks into a ribbon in the pericentriolar position is fascinating. What is the role of this organization and why should anyone care? Decades have been spent understanding the mechanism and the physiological significance of cisternal stacking and there is little in the way of a concrete answer. These membranes undergo extensive fragmentation at the onset of mitosis and for decades it has been proposed that this is necessary for partitioning of the membranes during cell division. But is it? A clear experiment to test this proposal is lacking. A newish proposal is that these membranes fragment to ensure spindle dynamics. There is a good reason to believe in this data because Golgi membranes have the capacity to nucleate microtubule asters, which could potentially cause havoc for a cell to build a bipolar spindle in metaphase. May be then Golgi fragmentation is way to prevent the interference in assembly of a bipolar spindle.

I like this work because it provides insights into the role of Golgi in spindle dynamics in association with mitotic events.

I don't have any issues with the technical aspects of the paper. There is a heavy use of pharmacology, but this is likely the best approach to prevent cells from adapting to alternatives in molecular approaches of knockdown or over expression of genes.

Answer: We thank the reviewer for appreciating our work.

1. What is CE? The authors throw this abbreviation without explaining anywhere what it defines.

Answer: We thank the reviewer for pointing out our oversight. The correct definition has been added in the text (page 3, line 55).

2. In cells that contain an intact Golgi and allowed to enter mitosis by affecting the activity of CDK1, would the cells enter mitosis upon treatment with a small dose of nocodazole? Nocodazole depolymerises microtubules and causes stacks of Golgi to separate and disperse in the cytoplasm. Does short term treatment with nocodazole activate Src?

Answer: This is a very interesting aspect that we have already addressed in a way that we think is sufficient to answer the question. In Barretta et al. 2016 (ref. 14), we demonstrated that rapid induction of Golgi ribbon disassembly into stacks by short treatment with BFA induces activation of Src at the Golgi and of Aurora. A comment regarding this point and a reference have been added to the text (page 3, line 76).

Regarding the second question we tested what is the effect of treating cells blocked with SP in G2 with low concentrations of nocodazole. We treated Hela cells with SP600125 for 3h and added 5 μ M Nocodazole during the last two h to depolymerize the microtubules. As shown in the attached figure, the treatment with nocodazole is not able to restore the mitotic index in the cells with compact Golgi treated with SP600125. The nocodazole is able to depolymerize microtubules, while SP600125 is able to compact the Golgi. We analyze the mitotic index in control cells and in cells treated with drugs. However, in our experiments we found that many of the cells treated with SP and Nocodazole underwent apoptosis. Thus, we think that our data in current form are too preliminary to be included in this manuscript. However, these findings resulted in an interesting opportunity that we could exploit to test if the combination of the two drugs could selectively kill cancer cells.

3. The authors are encouraged to include a paragraph on why this system is important for a subset of cells. Yeast and fully polarised and terminally differentiated cells do not organise their Golgi in the form seen in fibroblasts and hence might not need this mechanistic control.

Answer: We think this is a crucial aspect at the base of the Golgi checkpoint. A short paragraph and a reference to a recent review discussing the issue have been added in the discussion section (page 19, line 499).

4. is there any way to know what comes first: dissolution of the focal adhesions or the separation on the Golgi ribbon into stacks?

Answer: This is an issue that we are currently addressing. Our working hypothesis, based on literature data (ref. 4, 5, 6) and discussed in a recent review (ref. 64) is that the G2-specific expression of the scaffold DEPDC1B induces Focal Adhesion disassembly, which acts as a signal to induce Golgi unlinking. In any case, this is a complex project in which, in the active phase of development, the preliminary data supports our hypothesis. Still, they are part of a different project, which will be developed in a new paper.

Otherwise, this is a fine piece of work that should be published.

Answer: We thank the reviewer again for her/his appreciation and consideration of our work.

Reviewer #2:

In this manuscript, the authors developed a novel strategy to interrogate the requirement for Golgi ribbon separation as a mitotic checkpoint. This strategy involves cell cycle synchronization, inhibition of JNK2 with SP600125 to prevent the Golgi unstacking in G2, and then activation of CDK1 using the Wee1 inhibitor MK1775. This new method is likely to be useful to others in the field. Here, the authors use this strategy to provide more evidence for the importance of Golgi unstacking as a mitotic checkpoint. Importantly, the authors show that when mitosis is induced with an intact Golgi there is an increase in mitotic defects, particularly spindle multipolarity and defective cytokinesis resulting in binucleation. Additional experiments suggested a mechanism for these defects is lack of recruitment of Aurora-A kinase at centrosomes.

The experiments are mostly convincing and well-controlled.

Answer: We thank the reviewer for her/his appreciation and consideration of our work.

A few points should be addressed by the authors:

1. The final piece of data (Figure 6D) regarding increased cellular transformation is potentially very interesting but not very compelling in the present form. It appears that NIH3T3 cells were treated one time with SP and/or MK and/or BFA and then they were washed out and cells were allowed to grow for 2 weeks to form foci. It is a bit difficult to understand how this one time treatment led would lead to increased foci formation over the course of 2 weeks. The authors may need to explain the assay and interpretation better. In addition, the foci numbers went from 10 in the control to 20 in the SP+MK

condition - the authors should show images of the crystal violet staining of representative plates so the reader can better visualize the assay.

Answer: The reviewer has highlighted an aspect with potentially significant implications from a functional point of view, but perhaps it is still developed at a preliminary level. Considering the difficulty of explaining the observed phenotype and similar comments made by the third reviewer, we decided that a good option at this level is to eliminate any reference to the transformation data and develop them more in-depth in a different manuscript.

2. The authors use different cells in different figures, including NRK, HeLa and TERT-RPE1. The authors should provide some rationale as to why these different cell lines are being used in different assays.

Answer: A similar concern has been expressed by reviewer #3. We understand their concerns, and here we repeat the same answer.

The use of different cell models is explained by the need to employ different experimental approaches to investigate the consequences of entering mitosis with an intact ribbon and carry out the proper controls. An important aspect is that all three cell models showed that entry into mitosis with intact Golgi results in a high percentage of aberrant spindles.

Regarding individual models, normal rat kidney (NRK) cells are non-cancerous and generally used as a mitosis model because they are easy to synchronize. However, we could not efficiently deplete GRASP55 in these cells to induce unlinking, and we could use only BFA to induce Golgi disassembly as a control to SP65100 and MK1775 treatments. For this reason, we looked for a cell line that could be efficiently transfected and synchronized and characterized by a low basal level of mitotic defects. Human retinal pigment epithelial-1 (RPE-1) cells satisfy these needs. They are increasingly used as a model to study mitosis because they represent a non-transformed alternative to cancer cell lines, such as cervical adenocarcinoma HeLa cells. Using RPE-1 cells, we efficiently reduced GRASP55 expression to induce constitutive unlinking as a control of our treatments.

A different experimental challenge has been represented by the need to precisely control the expression levels of Aurora A expression in the experiment focused on addressing the role of this kinase. Based on our previous experience (Persico et al. 2010), it is fundamental to express Aurora A levels that are similar to the endogenous to avoid the spindle defects caused by Aurora-A overexpression. An additional crucial factor is the efficiency of transfection in the cell population. We have been able to find the correct experimental condition only using HeLa cells, which are a largely employed model for cell cycle studies. However, they have a higher baseline fraction of cells with aberrant spindles.

Importantly, in all the cell models examined, the forced entry into mitosis with an intact ribbon causes spindle aberrations, indicating the generality of the process.

Short comments to the use of the cells have been introduced in the text (page 6 line 145, page 8 line 222, page 11 line 303).

3. Please describe in the methods section how the mitotic index assay is performed.

Answer: We thank the reviewer for highlighting this shortcoming. A more detailed description has been added in the materials and methods section (page 19, line 543), and the control of synchronization efficiency using FACS Analysis is now shown as supplementary figures (Supplementary S1 and S3).

4. Please check and define all abbreviations. It was not clear that some abbreviations, such as CE and SAC, were defined for the reader.

Answer: As suggested by the reviewer, a precise definition of the abbreviations has been added in the text. CE as centrosome (page 3, line 55) and SAC as Spindle Assembly Checkpoint (page 8, line 201).

Reviewer #3 (Comments to the Authors (Required)):

In this paper, Mascanzoni et al. investigate a possible implication of Golgi ribbon cleavage in mitotic spindle formation and cytokinesis. Previous work from the Colanzi's group reported that the TGN-associated kinase Src is activated at G2/M by Golgi unlinking, and that this activated Src phosphorylates mitotic Aurora-A kinase, thus promoting its recruitment to the centrosome and its activity. Subsequent CDK1 activation triggers mitotic entry. They also described that JNK2 has a crucial role in Golgi cleavage during G2 through phosphorylation of the Golgi protein GRASP65 on Ser 277. Based on these results, they now combine JNK2 inhibition (to block Golgi unlinking) with CDK1 activation (to stimulate mitosis entry) and analyse subsequent perturbations in cell division. They conclude that Golgi cleavage-induced Aurora-A signalling is crucial for correct spindle formation and cytokinesis.

This work tackles a very interesting question: how mitotic events at different subcellular structures are regulated and coordinated in order to achieve a successful cell division. The authors conclude that Golgi ribbon cleavage acts a crucial player, not only at the G2/M checkpoint as previously reported, but it is also pivotal for proper spindle formation and faithful cell division and cytokinesis.

My major concern on the manuscript is the extensive use of kinase inhibitors as the main (almost the only) strategy to support conclusions. Both inhibition of JNK2 and stimulation of CDK1 might produce unrelated effect that might conceivably affect the interpretation of the results. To rule out this possibility, the authors employ BFA treatment. However, in my opinion the mere employment of BFA, although suitable, is not sufficient per se to demonstrate that the observed phenotypes are exclusively due to Golgi unlinking.

In this regard, some Golgi proteins, such as golgin160, the receptor of the dynein/dynactin complex at the GA, strongly associate with the centrosome in the presence of BFA (Yadav et al., 2012). It is reasonable to believe that relocation of Golgi proteins after BFA treatment might alter centrosome activity. In fact, Figure 2C and 4B of this work show that BFA alone reduces Aurora A intensity at the centrosome and produces aberrant spindles, respectively.

Answer: In order to demonstrate that the spindle and division defects are correlated to Golgi unlinking, in addition to Brefeldin, we also employed the depletion of GRASP55, a Golgi matrix protein that drives the ribbon formation. The depletion of GRASP55 induces a constitutive and irreversible unlinking of GC into isolated stacks and can rescue a correct spindle formation, confirming that the formation of aberrant spindles observed is the direct consequence of the lack of Golgi unlinking during G2. For details, see page 8 (line 219).

Moreover, even if the use of Brefeldin can induce minor defects of spindle formation (about 19% compared to control, 10%; Fig. 2 E), we underline the conspicuous rescue of correct spindle formation in cells treated with BFA+SP+MK compared to SP+MK treated cells.

The authors reported in their previous paper Cervigni et al. 2015 that a single point mutation in GRASP65 (Ser277 phospho-mutant) recapitulates the effects on the Golgi structure observed after JNK2 inhibition. Therefore, the most relevant data of this work, notably those concerning to the centrosome and spindle poles, should be confirmed by using a similar approach.

Answer: In principle, we agree with the reviewer's comment. However, the proposed experiment is extremely complex and, unfortunately, characterized by the fact that the expression of the mutant has a limited effect on the transition between mitosis, as already shown in Cervigni et al., 2015. An essential aspect of the experimental approaches was the possibility of inducing a rapid and efficient perturbation of the Golgi unlinking process during G2.

In this regard, we have tried many alternative methods to induce a rapid and synchronous block of Golgi fragmentation. However, the only method that has proven to be suitable is the administration of JNK inhibitors, which, in a few hours, can induce a strong and rapid block of Golgi unlinking and entry into mitosis. For this reason, we have dedicated many experiments to demonstrate the direct correlation between Golgi unlinking and spindle formation by artificial disruption of the Golgi with Brefeldin or GRASP55 depletion. Moreover, considering the extent of the effects observed and the significance of recoveries of the correct phenotype (i.e., spindle formation) after Golgi disassembly, we think that the role of Golgi unlinking in forming the spindle is adequately demonstrated.

Furthermore, in order to strengthen the conclusions drawn, appropriate control should be included in most of the experiments performed as detailed below.

Major points:

- o Why different cell lines are employed throughout the paper. NRK and HeLa are transformed cells, whereas RPE1 are primary immortalised cells. Both Golgi and centrosome composition and activity greatly vary among cell lines. For instance, the Golgi of RPE1 cells actively nucleates microtubules, while in Hela cells this activity is highly variable or absent.

Answer: A similar concern has been expressed by reviewer #2. We understand their concerns, and here we repeat the same answer.

The use of different cell models is explained by the need to employ different experimental approaches to investigate the consequences of entering mitosis with an intact ribbon and carry out the proper controls. An important aspect is that all three cell models showed that entry into mitosis with intact Golgi results in a high percentage of aberrant spindles.

Regarding individual models, normal rat kidney cells are non-cancerous and generally used as a mitosis model because they are easy to synchronize. However, we could not efficiently deplete GRASP55 in these cells to induce unlinking, and we could use only BFA to induce Golgi disassembly as a control to SP65100 and MK1775 treatments. For this reason, we looked for a cell line that could be efficiently transfected and synchronized and characterized by a low basal level of mitotic defects. Human retinal

pigment epithelial-1 (RPE-1) cells satisfy these needs. They are increasingly used as a model to study mitosis because they represent a non-transformed alternative to cancer cell lines, such as cervical adenocarcinoma HeLa cells. Using RPE-1 cells, we efficiently reduced GRASP55 expression to induce constitutive unlinking as a control of our treatments.

A different experimental challenge has been represented by the need to precisely control the expression levels of Aurora A expression in the experiment focused on addressing the role of this kinase. Based on our previous experience (Persico et al. 2010), it is fundamental to express Aurora A levels that are similar to the endogenous to avoid the spindle defects caused by Aurora-A overexpression. An additional crucial factor is the efficiency of transfection in the cell population. We have been able to find the correct experimental condition only using HeLa cells, which are a largely employed model for cell cycle studies. However, they have a higher baseline fraction of cells with aberrant spindles.

Importantly, in all the cell models examined, the forced entry into mitosis with an intact ribbon causes spindle aberrations, indicating the generality of the process.

Short comments to the use of the cells have been introduced in the text (page 6 line 145, page 8 line 222, page 11 line 303).

Furthermore, the authors employ different synchronization protocols depending on the cell type used. Since most of the experiments performed rely on an accurate G2 arrest and subsequent mitotic entry, it would be convenient to include a simple flow cytometry analysis resulting from these synchronisation methods.

Answer: As suggested by the reviewer, synchronization methods are introduced in the methods section (page 19, line 535), and data from FACS analyses for NRK cells and RPE-1 cells have now been added as supplementary material (Supplementary S1 and S3). For HeLa cells, the FACS analysis is reported by Barretta et al. 2016 (Supplementary Figure 2).

Figure 1.- The authors claim that the Golgi ribbon remains intact in the presence of kinase inhibitor SP600125 and in the presence of both SP600125 and MK1775. However, I would argue that a number of Golgi objects can be appreciated in both the top and bottom panels of Figure 1B. It would be interesting to quantify the number and area of the Golgi elements present in all these conditions and compare it with those of Golgi from G1 cells. This would provide a robust assay to evaluate the efficiency of the treatments employed, and show that the variability observed in Golgi elements is not greater than what normally observed in intact Golgi in NRK cells. Indeed, one would expect that values of G1 and G2-treated cells would be similar and lower than those of control G2 cells. In addition, it would be good to indicate how G2 cells were identified in each condition.

Answer: As suggested by the reviewer, a quantification of Golgi elements in all the conditions shown has been added to Figure 1 (Figure 1C) and commented on in the text (page 6, line 163).

As previously indicated (Persico et al. 2010), cells in G2 are identified by the presence of non-condensed chromosomes and separated centrosomes.

o Figure 2. Please explain what is the difference between cells showed in Figure 1B SP+MK and those showed in Figure 2D SP+MK. In the first case, the authors conclude that under these conditions the Golgi

ribbon remained intact, while in the second one they state that the GA is highly fragmented probably due to CDK1 stimulation of Golgi disassembly pathways distinct from the JNK2-mediated GRASP65 phosphorylation. Accordingly, BFA has no effect on the area of Golgi fragments induced by SP+MK treatment. This issue has to be clarified.

Answer: The extent of mitotic Golgi fragmentation was determined by measuring the size of mitotic clusters in control cells or cells treated with SP and MK in the absence or presence of BFA. The results indicate that after entry into mitosis, the Golgi complex is fragmented also in the cells that had an intact ribbon in G2 (treated with SP65100). Moreover, the addition of Brefeldin does not change the fragmentation levels observed in prometaphase in the cells treated with SP. Overall, these data indicate that the defects of spindle formation are not the consequence of the persistence of large Golgi clusters in prometaphase but are exclusively the direct consequence of the lack of Golgi unlinking during the G2 phase. Moreover, our data exclude the possibility that Brefeldin rescues spindle formation by disassembling big Golgi fragments in prometaphase.

Figure 2C. Representation of data is tricky and impairs the interpretation of the relevance of aberrant mitotic spindle in the different conditions. For instance, the number of aberrant spindles is higher in SP treated cells (that are blocked in G2) or in BFA treated cells (that enter mitosis with a disassembled GA) than in control cells. In order to appreciate the relevance of this experiment, I would suggest to report the percentage of aberrant spindles to the total spindles number showed in Figure 1C. Also, statistical significance of control versus control BFA treated cells and control BFA versus BFA+SP+MK should be calculated and plotted.

Answer: Figure 2C reports the percentage of aberrant spindles with respect to the total spindles number. In this Figure, the difference of aberrant spindles between the control and Brefeldin is not significant, as the P value is 0.2978 (ns). Similarly, the difference in aberrant spindles between the control cells treated with BFA and BFA+SP+MK is not significant. The P value is 0.2750 (t-test), as added in Figure legend (Figure 2C).

Conversely, the addition of BFA to SP+MK-treated cells induced a substantial and significant recovery. Finally, also SP addition alone does not induce a significant effect.

o Figure 3. The experiments included in this figure are highly relevant to support the conclusions drawn from this work because it makes use of a different approach and employs a non-transformed cell line. Therefore, it is important to well characterize the behaviour of the GC after drug treatments in parallel with the number of aberrant spindles. The degree of Golgi fragmentation in control RPE1 and after GRASP55 KD under all the conditions tested has to be shown and quantified as before. Furthermore, cell synchronisation efficiency should be monitored through FACS analysis and reported. In my experience, double thymidine block is not the best method to synchronise RPE1 cells. Does SP treatment on its own lead to aberrant spindle formation in GRASP55 KD cells? How do the authors explain the high number of aberrant spindles in SP+MK treated GRASP55 KD cells?

Answer: As suggested by the reviewer, a quantification of Golgi fragmentation after drug treatments has been added to Figure 3 (new Figure 3A) and commented on in the text (page 9, line 235). The control of synchronization efficiency with the double treatment of thymidine in RPE-1, using FACS Analysis, is

now shown in Supplementary Figure S3. As reported by the reviewer, SP600125 alone induces a minor fraction of aberrant spindles, as reported in ref. 55. The residual number of aberrant spindles in GRASP55-depleted cells treated with SP+MK probably may depend on Golgi independent effects of SP or on incomplete GC unlinking. In any case, the recovery effect after depletion is notable in magnitude and highly significant.

Figure 3A simply shows the efficiency of GRASP55 silencing and it is not directly relevant to the main conclusions of the manuscript. Therefore, it is highly recommended to move it to the supplemental material.

Answer: The Figure 3A has been moved in supplemental material, named as Figure Supplementary S4.

Figure 3B: Same comment as for Figure 2C regarding data representation

Answer: in the Figure 3B the percentage of aberrant spindles is reported as relative to the total spindles number.

o Figure 4. Please show Golgi labelling in Figure 4A to estimate the degree of Golgi unlinking under these conditions. The dispersion of data points in Fig 4B and 4C is very high. How were G2 and prophase cells chosen for this analysis? Since PCNT is affected by SP+MK treatment (Figures 4 and 5), a centriolar marker as ninein may be more appropriate for this analysis.

Answer: We have shown the Golgi labelling in Figure 4A, as suggested by the reviewer.

The dispersion of data points is normal because, in each cell, the position of the centrosome is stochastic in G2 and prophase. Moreover, the wide dispersion of distance measurement values between centrosomes is a consequence of the complex choreography of movements that centrosomes show during G2/M, starting from their separation, movement outside the nucleus and rotation, as recently shown by a reported publication (Frye et al. 2020). The cells in G2 are selected by identifying cells where the centrosomes are separated, and the DNA is not condensed. The cells in prophase are selected for having separated centrosomes and condensed chromatin.

Regarding the G2 phase, we found that pericentrin is a reliable marker to recapitulate the centrosome's position and behaviour. Instead, we agree with the doubts expressed by the reviewer regarding mitosis, as MTOC fragmentation may occur in this case, which, however, does not concern this specific figure.

o Figure 5. In this figure, another very relevant result sustaining the conclusions of the work is shown. SP+MK treatment reduces accumulation of Aurora A at the centrosome resulting in a significant increase in aberrant spindle percentage (from 20% to 80%). This effect was partially reverted by the ectopic expression of Aurora A. Therefore, the authors conclude that "a major cause of the GC-dependent spindle alterations is an insufficient level of Aurora-A activation at the CEs during G2/prophase...." However, SP+MK treatment seems to inhibit centrosome maturation, as can be appreciated through PCNT labelling of cells treated with SP+MK (shown in both figures 4A and 5A). Critical functions of Aurora A-kinase at the mitotic centrosomes are essentially mediated by the interaction with its cofactor Cep192. Since Cep192 accumulates at the PCM during centrosome maturation in G2, and this process is blocked by SP+MK treatment, it is difficult to distinguish whether the decrease in Aurora A recruitment is due to lack of Golgi unlinking-induced signalling or to inhibition of PCM maturation.

Answer: We thank the reviewer for identifying this potentially ambiguous aspect. The reviewer's conclusion is based on the fluorescence associated with the centrosomes in the current figure. The apparent reduction in fluorescence is probably only a consequence of the chosen image. In fact, by quantifying the fluorescence of pericentrin associated with centrosomes during G2, we found that SP treatment with MK had no effects, as shown in Supplementary Figure S5. Thus, the centrosome-associated fluorescence of Aurora-A shown in the figure does not depend on reduced centrosome maturation during this specific phase but only on reduced Aurora recruitment. In addition, we also would like to point out that the Golgi-dependent effects of Aurora on centrosome maturation shown by us in previous manuscripts refer to later stages of mitosis and were measured during prometaphase and metaphase. Our experiments show that in G2/M transition, the fluorescence of pericentrin in the control cells is similar to the fluorescence of pericentrin of cells treated with SP+MK, indicating that this treatment directly affects Aurora-A recruitment, and it is not an indirect effect of the presence of a reduced centrosomal matrix. For details see pages 11-12.

Given the complexity of the mechanisms underlying recruitment and activation of Aurora A at the centrosome, data obtained should be further confirmed by depleting GRASP55 or GRASP65 or, even better, by expressing a constitutively inactive version of GRASP65 (Ser277 phospho-mutant).

Answer: The experiment proposed by the reviewer is, in principle, highly appropriate but, unfortunately, not applicable in this case. The proposed experiment is extremely complex and, unfortunately, characterized by the fact that the expression of the mutant has a limited effect on the transition between mitosis, as already shown in the manuscript of Cervigni et al. 2015.

In my opinion, Figure S2 showing PCM fragmentation after SP+MK treatment and causing multipolar spindles should be included in the main text.

Answer: The Figure S2 is moved in the main text such as new Figure 5.

Figure 6. From my point of view, results shown in this figure are too preliminary and difficult to integrate with the rest of data. A deeper characterization is required to propose that Golgi unlinking has a role in midbody cleavage, especially taking into account that at this stage of mitosis, the Golgi ribbon has already reformed in control cells.

Answer: We agree with the reviewer that this aspect needs to be clarified in greater depth. According to literature data, defects in the bipolar spindle formation can cause cytokinesis failure, thus providing an explanation to our observations. Thus, a reference to this aspect has now been added in the text (page 16, line 465) (Ganem N J et al., 2009; Godinho S A et al., 2009).

In the same way, there are not enough data to support that "alterations of the spindle and the tetraploidization caused by the lack of GC disassembly in G2 can potentially induce cell transformation".

Answer: As shown by others, an important and immediate consequence of cytokinesis failure is the formation of a tetraploid daughter cell with two centrosomes (Lacroix and Maddox, 2012). The tetraploid state is unstable and often develops into aneuploidy, finally leading to transformation.

However, our experiments on foci formation and transformation are preliminary, and, as also suggested by the second reviewer, we decided to eliminate our data on transformation and develop this point in a

new manuscript. At the same time, we can propose this intriguing point as a topic of discussion in a speculative manner, given that there is abundant literature data that connects the failure of cytokinesis to tetraploidization.

Figure 6A shows that SP treated cells are able to progress into mitosis. In that case, what is the advantage of adding MK inhibitor?

Answer: The reviewer is right. Even in presence of SP a fraction of the cells can enter mitosis. This also could offer an explanation for the fraction of aberrant spindles seen after SP treatment that is shown in Figure 3A. However, the addition of MK is an advantage in terms of efficiency, synchrony and number of cells starting the division process.

Why aren't mitotic indexes and the rest of phenotypes analysed at a longer SP treatment timepoint?

Answer: in Figure 6A (now 7A), we analyzed the cells fixed 24h after aphidicolin washout. This is a time sufficient to complete one round of cell division and thus, we observe the direct consequences on the completion of cytokinesis of a single round of mitosis. Conversely, in the experiments shown in Figure 3A, we used MK1775 as an inductor of entry in mitosis and fixed the cells at the mitotic peak to investigate the spindle organization.

Actually, 1 hour after adding MK inhibitor, only 20% of control cells were in metaphase.

Answer: Probably the reviewer is referring to the current supplementary Figure S6B. We think this is not a low number, as it corresponds to the fraction of cells that are expected to be in metaphase during mitosis, also in control cells. Probably, the cells treated with MK alone enter mitosis faster but without major alterations of the speed of progression through the various mitotic stages. Conversely, the treatment with SP and MK results in a slowing down of mitotic progression, as suggested by the increased fraction of cells in metaphase. We thank the reviewer for these interesting considerations that we will develop into more details as a different project.

The authors should propose an explanation about the high number of binucleate cells in SP+MK treated GRASP55 KD cells knowing that MK does not affect mitotic exit. Does SP alone trigger binucleation in KD cells? How does this compare to control cells treated with SP alone? Significance of data SP versus SP+MK in KD cells should be included.

Answer: SP600125 is able to induce minor aberrant effects in cells (Mili et al. 2016), giving origin to a minor fraction of binucleated cells; for this reason, also in GRASP55 KD cells treated with SP+MK, there is a fraction of binucleated cells. In addition, part of the residual effect could depend on a not completely efficient depletion of GRASP55.

Minor

points

- In the first paragraph of the introduction, the authors must state what CE stands for, or alternatively, use centrosome throughout the paper.

Answer: The correct definition has been added in the text (page 3, line 55).

- The graphs of all figures expressing percentages should have the same y-axis, possibly set to 100%.

Answer: we tried to modify the figures according to the reviewer's suggestions. However, we found that with a scale reported at 100%, the presentation of the data was less effective. Therefore, we would prefer to maintain the current format of the figures.

- Numbers referring to the quantification provided in the graphs (i.e mitotic index observed in control cells (20% {plus minus} 5%) ...) should be included in the text.

Answer: we have reported in the text the quantification of the graphs as required. However, in some sentences, we have not consistently reported all the data so as not to make the reading of the text more difficult. (page 6, lines 161,165 - page 7 lines 166, 182, 194 – page 9 lines 230, 231, 232, 234, 238 – page 12 lines 328, 389, 330, page 13 line 375- page 14 line 377).

February 22, 2024

RE: Life Science Alliance Manuscript #LSA-2023-02469-TR

Dr. Antonino Colanzi
Institute of Experimental Endocrinology and Oncology "G. Salvatore" (IEOS)
National Research Council
Via Pietro Castellino 111
Naples 80131
Italy

Dear Dr. Colanzi,

Thank you for submitting your revised manuscript entitled "The Golgi checkpoint: Golgi unlinking during G2 is necessary for spindle formation and cytokinesis". We would be happy to publish your paper in Life Science Alliance pending final revisions necessary to meet our formatting guidelines.

- please be sure that the authorship listing and order is correct
- please upload all figure files as individual ones, including the supplementary figure files; all figure legends should only appear in the main manuscript file
- please add ORCID ID for the secondary corresponding author -- they should have received instructions on how to do so
- please add the Twitter handle of your host institute/organization as well as your own or/and one of the authors in our system
- please add an Author Contributions section to your main manuscript text
- please add a conflict of interest statement to your main manuscript text
- please add your main, supplementary figure, and table legends to the main manuscript text after the references section
- there is a callout for Fig. 6D, and the figure has only panels A-C -- please correct

Figure Checks:

- please add scale bar to Figure 1B

A. FINAL FILES:

B. MANUSCRIPT ORGANIZATION AND FORMATTING:

Sincerely,

February 28, 2024

RE: Life Science Alliance Manuscript #LSA-2023-02469-TRR

Dr. Antonino Colanzi
Institute of Experimental Endocrinology and Oncology "G. Salvatore" (IEOS)
National Research Council
Via Pietro Castellino 111
Naples 80131
Italy

Dear Dr. Colanzi,

Thank you for submitting your Research Article entitled "The Golgi checkpoint: Golgi unlinking during G2 is necessary for spindle formation and cytokinesis". It is a pleasure to let you know that your manuscript is now accepted for publication in Life Science Alliance. Congratulations on this interesting work.

DISTRIBUTION OF MATERIALS:

Again, congratulations on a very nice paper. I hope you found the review process to be constructive and are pleased with how the manuscript was handled editorially. We look forward to future exciting submissions from your lab.

Sincerely,
